# Wasserstein Distances, Neuronal Entanglement, and Sparsity

**Shashata Sawmya**[1*], **Linghao Kong**[1*], **Ilia Markov**[2], **Dan Alistarh**[2,3,4], **& Nir Shavit**[1,3,4]
[1]MIT    [2]IST Austria    [3]Neural Magic    [4]Red Hat
{shashata, linghao, shanir}@mit.edu,
{ilia.markov, dan.alistarh}@ist.ac.at

## Abstract

Disentangling polysemantic neurons is at the core of many current approaches to interpretability of large language models. Here we attempt to study how disentanglement can be used to understand performance, particularly under weight sparsity, a leading post-training optimization technique. We suggest a novel measure for estimating neuronal entanglement: the Wasserstein distance of a neuron's output distribution to a Gaussian. Moreover, we show the existence of a small number of highly entangled "Wasserstein Neurons" in each linear layer of an LLM, characterized by their highly non-Gaussian output distributions, their role in mapping similar inputs to dissimilar outputs, and their significant impact on model accuracy. To study these phenomena, we propose a new experimental framework for disentangling polysemantic neurons. Our framework separates each layer's inputs to create a mixture of experts where each neuron's output is computed by a mixture of neurons of lower Wasserstein distance, each better at maintaining accuracy when sparsified without retraining. We provide strong evidence that this is because the mixture of sparse experts is effectively disentangling the input-output relationship of individual neurons, in particular the difficult Wasserstein neurons.

## 1 Introduction

Disentangling polysemantic neurons into their component, human-understandable features has been a longstanding goal of machine learning interpretability research (Olah et al., 2020; Jermyn et al., 2022; Elhage et al., 2022; Gurnee et al., 2023; Templeton, 2024; Gurnee et al., 2024). While neurons are the basic building blocks of neural network architectures, they do not map one-to-one with specific features. Instead, neurons frequently engage in polysemantic representations, where they are activated by multiple, unrelated concepts and detect diverse features (Arora et al., 2018; Mu & Andreas, 2020). It is suspected that every neuron is polysemantic to some degree (Lecomte et al., 2023), and so we will refer to all neurons as polysemantic in this work.

Due to the importance of highly polysemantic neurons in a network's computation (Bricken et al., 2023), the question of whether these neurons require more parameters naturally arises. However, the effects of polysemanticity on network performance under weight sparsity has not been well explored. Weight sparsification (Hoefler et al., 2021) aims to reduce the number of executed parameters in large language models (LLMs) by setting certain weight values to zero to improve efficiency. Various sparsification algorithms have been developed for this process (Han et al., 2015; Sun et al., 2023; Frantar & Alistarh, 2023). This paper investigates the relationship between an individual neuron's degree of entanglement (which we will formally define in a later section) and its ability to be sparsified in real-world models. To the best of our knowledge, this is the first work to explore this crucial perspective of entanglement-dependent model sparsification.

To better understand the impact of entanglement on sparsification, we introduce a novel metric that quantifies a neuron's degree of entanglement. This metric is the Wasserstein distance between a

---

[*]Equal contribution. Author order determined by coin toss.
[1]Code available at https://github.com/Shavit-Lab/Sparse-Expansion.

neuron's output distribution and a Gaussian (Equation 1). We find that neurons with a particularly high Wasserstein distance (Figure 1d, A8d) are crucial for the performance of a network and very sensitive to pruning. We provide evidence that a neuron's Wasserstein distance is related to its ability to distinguish similar inputs to different outputs through its dot product, and we refer to these neurons as especially entangled (Equation 2). Akin to previous works investigating special types of neurons (Gurnee et al., 2023; Stolfo et al., 2024; Gurnee et al., 2024), this work explores the role of crucial neurons with implications for interpretability, specifically in the context of network sparsity.

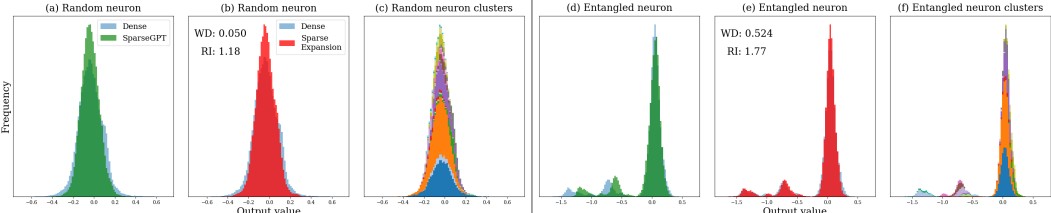

Figure 1: The output distributions of neurons in Llama-2-7B computed densely and at 90% sparsity on Wikitext-2. WD refers to the Wasserstein distance of the output distribution to a Gaussian. RI refers to the relative improvement of Sparse Expansion over SparseGPT. (a) The dense output distribution of a random neuron with a WD of 0.050 is well captured by SparseGPT, and (b) expanding this neuron via Sparse Expansion imparts only a small (18%) increase in performance. (c) The cluster outputs are all concentrated in close proximity to each other. (d) SparseGPT struggles to capture the dense distribution of an entangled neuron with a WD of 0.524. (e) Following expansion, the sparse output of the entangled neuron is much better captured, leading to more improvement (77%). (f) Each expert specializes over a different portion of the distribution.

To analyze the phenomenon of neuronal superposition under sparsity in greater detail, we create an experimental framework, which we dub Sparse Expansion. It expands a model into a mixture of sparse experts by clustering input embeddings layer-wise. Based on this clustering, Sparse Expansion utilizes the input-aware nature of the SparseGPT (Frantar & Alistarh, 2023) pruning algorithm to specialize different sparse experts to different sets of inputs, starting from the same base weights. Through Sparse Expansion, we are able to analyze the entangled neurons in much more detail, since now different subgroups of the inputs are being computed with different edges (Figure 1f, A8f). We find that as a neuron lose edges, its output distribution tends to shift toward a Gaussian distribution (Figure A9). However, through Sparse Expansion, the original output distribution can be better preserved under sparse computation (Figure 1e, A8e). We relate our findings to recent theoretical work on the bounds of neural computation under superposition (Hänni et al., 2024; Adler & Shavit, 2024).

Our main technical contribution is a detailed study of how model accuracy under sparsity is related to its degree of neuronal entanglement. In every LLM, there exist neurons that have striking, irregular output distributions (Figure 2c, A1). These neurons have an outsized effect on model performance and seem to be responsible for differentiating similar input vectors (Figure 2). We believe that the existence of these neurons is a manifestation of polysemanticity in real-world language models. We find that the Wasserstein distance to a Gaussian is a strong indicator of such neurons.

In the next section we explain such "Wasserstein neurons", neuronal entanglement, and the implication of ablating Wasserstein neurons in LLMs in detail. We then formulate our experimental framework Sparse Expansion and show how to effectively disentangle the input-output relationship of neurons through Sparse Expansion, as well as some empirical computational bounds. Finally, we present some results showing its performance relative to other state-of-the-art one-shot compression techniques in the hopes of inspiring future sparsification algorithms.

## 2 WASSERSTEIN NEURONS

### 2.1 CHARACTERIZING NON-GAUSSIAN NEURONAL OUTPUT DISTRIBUTIONS

We investigate the output distributions of individual neurons in all linear layers of transformer feed-forward networks (FFNs) during inference. Specifically, consider a linear operation $Y = WX + b$, where $Y \in \mathbb{R}^{n \times s}$ is the output matrix, $W \in \mathbb{R}^{n \times m}$ is the weight matrix, $b \in \mathbb{R}^n$ is the bias vector,

broadcasted across all neurons, and $\boldsymbol{X} \in \mathbb{R}^{m \times s}$ is the input matrix, where each column represents an input vector. Each neuron is an individual row of $\boldsymbol{W}$, and we collect individual scalar elements from the corresponding row in $\boldsymbol{Y}$ as the output distribution for that neuron.

We focus our analysis in Pythia-1.4B (Biderman et al., 2023), Llama-2-7B (Touvron et al., 2023), and Llama-3-8B (Dubey et al., 2024). Most neurons exhibit a reasonably Gaussian output distribution after their dot product with the input vector (Figure 1a, 2a). However, we find the existence of a small group neurons with highly non-Gaussian outputs (Figure 1d, 2c) in all FFNs (Figure A1).

To characterize the degree of difference in terms of the shape of these distributions—the non-Gaussian output distributions of certain neurons with the Gaussian-like output distribution of most neurons—we considered several metrics, such as entropy. However, the Wasserstein distance (WD) (Kantorovich, 2006; Villani et al., 2009) proved to be the most effective metric for quantifying this difference. In optimal transport theory, the WD measures the minimal transportation cost between two distributions, taking their geometry in real space into account.

To find the WD of every neuron to the Gaussian $\mathcal{N}$, we crucially first normalize the output distributions of each neuron $n$ to have zero mean and unit variance, and compare this normalized distribution $n'$ to $\mathcal{N}(0, 1)$. This normalization is performed because the range of neuron output distributions is quite variable, and we wanted to prioritize the differences in the shape of the distributions, rather than other properties. We use the 1-Wasserstein distance in one dimension, as shown in Equation 1.

$$W_1(n', \mathcal{N}) = \int_0^1 |F^{-1}(z) - \varphi^{-1}(z)| dz. \tag{1}$$

$F^{-1}$ and $\varphi^{-1}$ are the inverse cumulative distribution function of $n'$ and $\mathcal{N}(0, 1)$, respectively, which can be approximated with empirical data. To compute the WD of every neuron efficiently, we use the `SciPy` implementation (Virtanen et al., 2020). When computing the difference metric in this way, we find that our originally observed neurons (Figure 1d, A8d) have been designated correctly with high WD to $\mathcal{N}$. We thus term these neurons "Wasserstein neurons." We also observe little overlap between neurons with high mean weight magnitudes and Wasserstein neurons (Figure A4a).

We additionally analyze Pythia-1.4B across its training, from network initialization to the final step. We find that Wasserstein neurons do not seem to receive more weight updates than other neurons (Figure A2a). Interestingly, we also find that Wasserstein neurons arise relatively early on in training, within 10-20 billion tokens (Figure A2b). This phenomenon is likely related to and a manifestation of other observations that fundamental model training dynamics rapidly stabilize, such as the rank of the gradient or the largest eigenvalue of the loss hessian (Gur-Ari et al., 2018; Zhao et al., 2024; Noci et al., 2024). We leave further investigations into this crucial training period to future work.

## 2.2 WASSERSTEIN NEURONS AND ENTANGLEMENT

Here, we define and study the notion of entanglement of these Wasserstein neurons in greater detail by positing a new avenue to investigate entanglement. According to superposition theory, as the number of features increases relative to the number of neurons, features are forced to become non-orthogonal in order to represent more of them, thus increasing entanglement (Elhage et al., 2022). Consider neurons that must attend to multiple of these features. As the number of features increases, and different features are forced to become more similar in direction, such neurons must still manage to distinguish between them. Therefore, in this context, neurons that are highly entangled have the task of differentiating between similar input vectors, and mapping them to different output values.

To mathematically explore this concept, we study the input-output (IO) relationship of individual neurons. We introduce the metric "mapping difficulty" (MD), which measures how often a neuron must generate dissimilar outputs from similar inputs through its dot product computation. The MD for a particular neuron, given its weights and a set of inputs, is calculated as follows (Equation 2):

$$\mathrm{MD}(\boldsymbol{w}, \mathbb{X}) = \operatorname*{mean}_{1 \leq i < j \leq n} \left\{ \left( \frac{||y_i - y_j||}{N_y} \right) \Big/ \left( \frac{||\boldsymbol{x}_i - \boldsymbol{x}_j||}{N_{\boldsymbol{x}}} \right) \right\}$$

$$\boldsymbol{x}_i, \boldsymbol{x}_j \in \mathbb{X}, \quad y_i = \boldsymbol{w} \cdot \boldsymbol{x}_i, \quad n = |\mathbb{X}|$$

$$N_{\boldsymbol{x}} = \operatorname*{max}_{1 \leq i < j \leq n} \{||\boldsymbol{x}_i - \boldsymbol{x}_j||\}, \quad N_y = \operatorname*{median}_{1 \leq i < j \leq n} \{||y_i - y_j||\} \tag{2}$$

$\boldsymbol{x}_i$ and $\boldsymbol{x}_j$ represent two distinct input vectors from the set of inputs $\mathbb{X}$. $y_i$ and $y_j$ represent the two output scalars as a result of the dot product of an individual neuron's weights $\boldsymbol{w}$ with the inputs. For every pair of inputs, we compute the $L^2$ norm of their difference, then scale the norms between zero and one using the maximum norm $N_{\boldsymbol{x}}$. We then compute the $L^2$ norm of the difference in their corresponding outputs, and normalize them with the median norm $N_y$. More details on the rationale behind the normalizing factors can be found in Appendix A.8. The MD of a neuron can thus be calculated as the average of the ratio between the normalized difference in outputs to the normalized difference in inputs. Intuitively, a greater MD means that a neuron generally increases the separation of similar inputs into more dissimilar outputs.

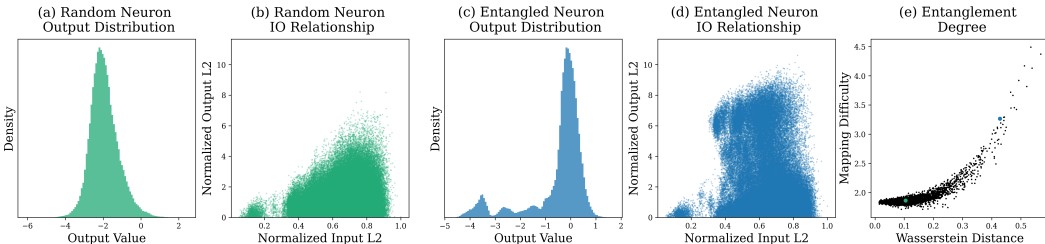

Figure 2: A measure of neuronal entanglement. (a) The output distribution of a random neuron. (b) The normalized $L^2$ plot of a random neuron's pairs of inputs and outputs. (c) The output distribution of a Wasserstein neuron. (d) The normalized $L^2$ plot of a Wasserstein neuron's pairs of inputs and outputs. This neuron must map fairly similar inputs to outputs that are very far apart through its dot product operation. The neurons are from the up projection matrix of the second FFN block in Pythia-1.4B. (e) The MD of a neuron is highly correlated with its WD. The selected random and Wasserstein neurons are highlighted in their respective colors.

For the two neurons we have selected before, we plot the normalized $L^2$ for pairs of inputs $(\|\boldsymbol{x}_i - \boldsymbol{x}_j\|/N_{\boldsymbol{x}})$ and outputs $(\|y_i - y_j\|/N_y)$, as defined in Equation 2. These inputs and outputs were collected over the course of running the Wikitext-2 dataset (Merity et al., 2016) through Pythia-1.4B. For the random neuron, as the difference between inputs decreases, so too does the difference between outputs (Figure 2b). However, for the Wasserstein neuron, this is not the case—even relatively similar inputs are mapped to outputs almost as far apart as the entire range of the neuron (Figure 2d). A clear trend between the MD of a neuron and its WD emerges (Figure 2e), and the two measures are highly correlated. Thus, we propose the WD of a neuron's output distribution to a Gaussian as a novel metric of entanglement, with Wasserstein neurons being particularly entangled.

## 2.3 Effect of high Wasserstein neurons on sparsification

In the previous section, we have related Wasserstein neurons to a novel formulation of entanglement. Now, we show that such neurons also have a substantially outsized effect on model performance under sparsity. In Llama-3-8B, if just 3% of all neurons—those with the highest WD—are sparsified via SparseGPT in every FFN, model performance significantly degrades. This degradation is far more severe than when 3% of random neurons are sparsified, and remains true when compared to sparsifying the same number of other important neurons, such as those with the greatest mean and variance in their output distributions and even those with the greatest mean weight magnitude. As compression increases, this effect becomes more obvious (Figure 3a). Therefore, Wasserstein neurons are crucial for maintaining accuracy and are severely limited in their ability to be compressed.

To better understand which specific capabilities are impacted by neuron entanglement, we evaluate the Llama-3-8B model with its Wasserstein neurons sparsified across several language model evaluation benchmarks. We select five tasks spanning four broad categories, similar to the original Llama-3 work (Dubey et al., 2024). For reading comprehension, we use the 1-shot variant of the SQuAD 2.0 dataset (Rajpurkar et al., 2018). To assess knowledge reasoning and mathematical capabilities, we evaluate the model on the 5-shot TriviaQA-Wiki (Joshi et al., 2017) and 5-shot GSM8K (Cobbe et al., 2021) datasets, respectively. Finally, to evaluate general reasoning, we test the model on two benchmarks: an easy task, 5-shot MMLU (Hendrycks et al., 2020), and a more challenging task, 3-shot Chain-of-Thought (CoT) Big Bench Hard (BBH) (Suzgun et al., 2022).

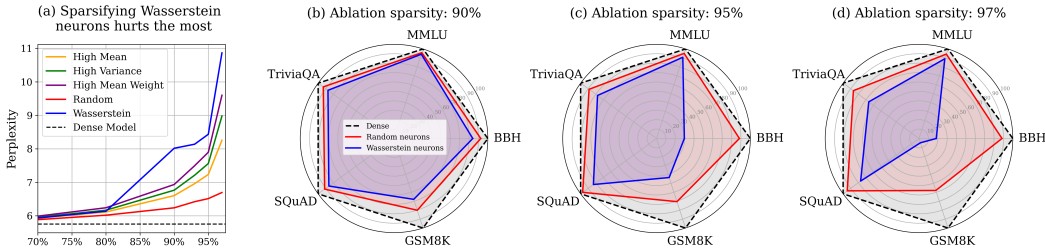

Figure 3: Entangled neurons are much more sensitive to compression. In Llama-3-8B, 3% of neurons from every FFN linear layer are sparsified via SparseGPT in an unstructured manner with a subset of the Wikitext-2 train dataset as calibration data. (a) Sparsifying Wasserstein neurons (blue) impairs the model more than sparsifying neurons with the highest output distribution means (orange) and variances (green), those with the highest average mean weight magnitude (purple), and considerably more than random neurons (red). Perplexity is measured on the Wikitext-2 test dataset. (b-d) Sparsifying the Wasserstein neurons (blue) affects general and mathematical reasoning much more than random neurons (red), as shown in the capability charts. At higher levels of neuron sparsity ($\geq 95\%$), ablating Wasserstein neurons leads to a collapse in model performance, which does not occur with random neurons.

Our findings reveal that when just a small fraction of neurons (the top 3% Wasserstein neurons) are sparsified, the model's performance on complex tasks involving general reasoning and mathematical understanding is significantly impacted. However, when the same level of sparsification is applied to random neurons, the model is able to preserve most of its capabilities effectively. Additionally, as a neuron is increasingly sparsified, the output distribution becomes more Gaussian (Figure A9, A10). This in turn places even more stress upon the neuron—not only is it contending with decreasing mean and variance of the output distribution (Figure A11), but also with the less expressive distribution shape. Thus, it seems that, especially at the higher sparsities that we are analyzing, the irregular shape of the entangled neurons is much more challenging to model with fewer weights than a Gaussian-like distribution. Furthermore, partially due to their slightly lower mean weight magnitudes (Figure A4a), Wasserstein neurons are actually sparsified more by SparseGPT during unstructured sparsity, compounding this issue (Figure A4b). However, keeping Wasserstein neurons dense at the cost of sparsifying all other neurons even more also does not seem to be the solution (Appendix A.7). To investigate the difficulty of sparsifying entangled neurons and the relationship between superposition and performance, we introduce Sparse Expansion.

## 3 AN EXPERIMENTAL FRAMEWORK TO STUDY DISENTANGLEMENT

To better study Wasserstein neurons and the phenomena between entanglement, sparsity, and performance that we observe, we create the experimental framework Sparse Expansion. It is inspired by recent work on the theoretical limits of computation within superposition (Hänni et al., 2024; Adler & Shavit, 2024). Sparse Expansion was designed to achieve two goals in real-world models. First, it must originate from a trained dense model and not be retrained. This way, the dynamics of a single neuron, in particular Wasserstein neurons, can still be studied in depth after the model has been expanded. Second, from a theoretical perspective, it must test how varying the number of effective features in the input affects the number of required weights. Therefore, the relationship between superposition and sparsity the can be further understood.

### 3.1 SPARSE EXPANSION IN DETAIL

Sparse Expansion clusters the inputs to each layer into separate groups via an optional PCA dimensionality reduction and K-means clustering. Each expert is then sparsified via the SparseGPT algorithm (Algorithm A1). Briefly, SparseGPT approximates the optimal sparse matrix of a layer with the Hessian of the error relative to the parameters of the layer $Y = WX + b$. Doing so yields $H = XX^T$, where $H$ is the Hessian matrix.

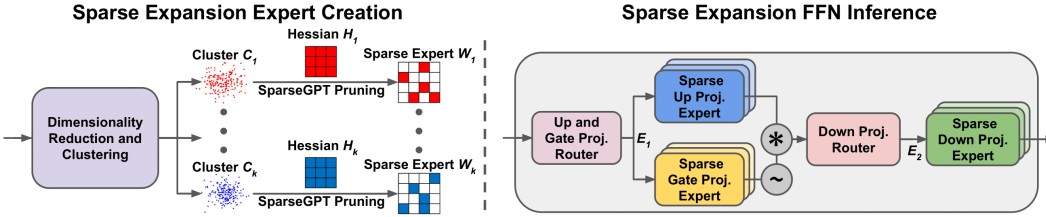

Figure 4: The Sparse Expansion process. One-shot expert creation process of Sparse Expansion (left). Inference process in a FFN of an expanded model (right).

During inference, each input will be passed through the PCA and K-means model to decide its expert, then routed to the corresponding expert for the matrix multiply (Algorithm A2). As the routing is done via K-means on a lower dimension, and the PCA is a very low dimension matrix multiply operation, both are inexpensive to add on to normal LLM inference. Furthermore, routing in this manner prevents the need to train and run a more expensive router.

Our design explicitly achieves the goals we set out. First, by starting from a dense model, we are able to study how the separation of inputs affects individual neurons, which we would not be able to do for the same neuron index across experts in a MoE model such as Mixtral (Jiang et al., 2024) and DeepSeek (Guo et al., 2025). Second, by utilizing SparseGPT, each expert has its weights sparsified and tailored to a subset of inputs, testing the theoretical limits of how many weights are necessary to model a given number of features.

## 3.2 SPARSE EXPANSION DISENTANGLES NEURONS

We revisit the output distributions of neurons to determine the effect that clustering has in a sparse setting. First, we repeat the sparsification experiment conducted in Figure 3 on Wikitext-2 in Llama-3-8B. Now, for just the neurons we pruned, we expand them into 16 experts and measure the recovery in performance. Sparse Expansion is able to recover significant performance following Wasserstein neuron sparsification, much more than it does during random neuron expansion. However, the recovery in performance for random neurons is not as noticeable, because these neurons were not under significant entanglement initially (Figure 5a). Furthermore, both the weighted cluster WD and weighted cluster MD of the majority of neurons decreases as a result of Sparse Expansion. The weighted cluster WD and MD are calculated as the average WD and MD within each cluster, weighted by cluster size. This is especially true for Wasserstein neurons, where 98% of neurons have a decrease in weighted WD by a median of 42% per neuron (Figure 5b), and where 96% of neurons have a decrease in weighted MD by a median of 9% per neuron (Figure 5c).

For Llama-2-7B (Figure 1) and Pythia-1.4B (Figure A8), both models and both neuron types—random and Wasserstein—improve through Sparse Expansion, with the entangled neuron showing greater improvement. Furthermore, for the random neuron and especially for the entangled neuron, the geometry of the sparse output distribution in Sparse Expansion much more closely matches that of the dense distribution.

We also provide a visualization for the specialization of each cluster. Figure 1 and Figure A8 each show the sparse output distributions of each individual cluster, with a different color per expert. For the randomly selected neurons, there is still an improvement, although each expert is for the most part responsible for approximately the same range and shape. For the entangled neurons, there is significant specialization for different parts of the distribution further away from the mode.

However, Sparse Expansion is not limited to improving just Wasserstein neurons. Across different sparsities and across different models, all but a tiny fraction of neurons improve through Sparse Expansion (Figure A12). Thus, like other work regarding polysemanticity (Lecomte et al., 2023; Bricken et al., 2023), we believe that, in fact, every neuron is to some extent in entanglement. However, Wasserstein neurons are the most obviously entangled ones, and they benefit more from sparse disentanglement, especially at higher sparsities. Finally, we note that other metrics of the dense neuronal output distribution, such as their means and variances, fail to act as a predictor of neuronal improvement to the degree that the Wasserstein distance to a Gaussian does (Figure 7,

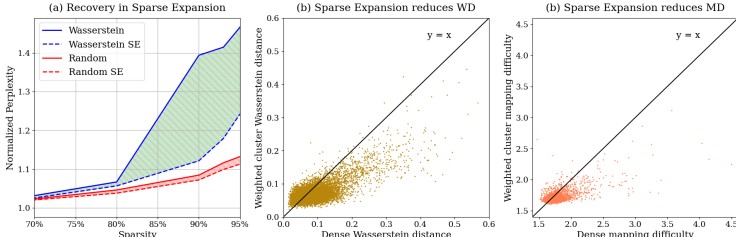

Figure 5: Sparse Expansion recovers performance of Wasserstein neurons. (a) Although Wasserstein neurons are penalized more under sparsity, they also recover better in Sparse Expansion compared to random neurons. We quantify this recovery using normalized perplexity relative to the dense model. Data from Llama-3-8B. (b) As a result of Sparse Expansion, the median decrease in WD per neuron is 19%. Although a few neurons with an initially low dense WD exhibit a higher average weighted WD, the majority (68%) of all neurons show a decrease in weighted WD. This is especially true in the top 10% of neurons with an originally high WD—the Wasserstein neurons. (c) Sparse Expansion also decreases the weighted MD by a median of 2% per neuron. 70% of all neurons and 96% of Wasserstein neurons show a decrease in weighted MD, the latter with a median decrease of 9% per neuron. (b, c) Data collected from of the up projection matrix in the second FFN of Pythia-1.4B.

A13). Thus, we believe that the WD to normal for a neuron's output distribution is a very suitable and intuitive metric of entanglement within a neuron.

### 3.3 MORE SPARSE EXPERTS BETTER FIT THE OUTPUT DISTRIBUTION

The complex dense output distribution of highly entangled neurons is difficult to model with a single sparse expert, as in the case of SparseGPT. In Figure 6, we show the output distribution of a Wasserstein neuron in both dense and sparse computation. As the number of sparse experts increases, the output distribution of the sparse computation more closely matches that of the dense computation, as measured in the WD between the two distributions. Furthermore, the relative improvement (RI) of Sparse Expansion over SparseGPT increases. In this paper, RI is measured as the ratio of the RMSE between the SparseGPT sparse computation and the dense computation, to the RMSE between the Sparse Expansion sparse computation and the dense computation.

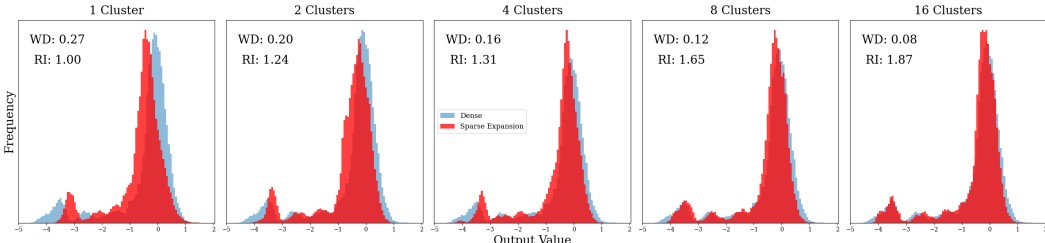

Figure 6: Modeling recovery with more experts. The sparse computation output distribution (red) better matches the dense one (blue) with more clusters. Sparsity is set to 90% for each expert. Here, WD refers to the Wasserstein distance between the Sparse Expansion sparse and dense output distributions, rather than to a Gaussian. RI represents relative improvement of Sparse Expansion ($n \geq 1$ clusters) over SparseGPT ($n = 1$ cluster). This is the same neuron from Figure 2c.

### 3.4 WASSERSTEIN DISTANCE BEST EXPLAINS IMPROVEMENT

So far, we have claimed that Wasserstein distance is not only a pertinent indicator of neuronal entanglement, but also a predictor of its improvement in Sparse Expansion over SparseGPT. To test this idea, we compare how well the RI is modeled by a neuron's output WD, mean output magnitude, and output variance. Of these metrics, a neuron's Wasserstein distance is most correlated with its improvement in sparse computation from disentanglement (Figure 7, A13).

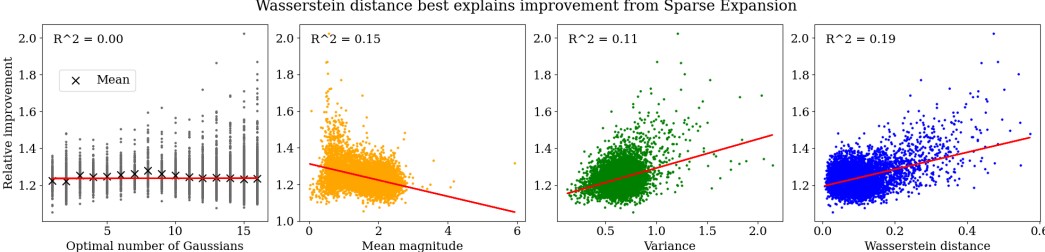

Figure 7: Wasserstein distance best explains improvement among tested metrics. The RI of each neuron in Pythia-1.4B was calculated as before and compared against the optimal number of Gaussians needed to model its output distribution (gray), the average magnitude of its output distribution (orange), the variance of its output distribution (green), and the Wasserstein distance of its output distribution to normal (blue). For each metric, the line of best fit is calculated, and the coefficient of determination $R^2$ is found. For each optimal number of Gaussians, the mean improvement is marked. Of these metrics, the Wasserstein distance best correlates with relative improvement. Data collected from of the second up projection layer in Pythia-1.4B.

In addition, we test whether the estimated number of components in a Gaussian mixture model (GMM) is enough to explain the improvement as a result of disentanglement. Specifically, given a neuronal output distribution, we applied Gaussian mixture modeling to determine the optimal number of Gaussians required to model the distribution, using the Bayesian Information Criterion (BIC) for evaluation. BIC is a metric that penalizes model complexity and tries to identify the minimum number of Gaussians which can optimally model the distribution. However, when testing the optimal number of Gaussians between one and sixteen models, our findings indicated almost no correlation ($R^2 \leq 0.001$) between the optimal number of Gaussians and the relative improvement in the Sparse Expansion setup, as seen in Figure 7. Thus, in our experiments, we find that the Wasserstein distance is a better indicator than others that we have tested.

## 3.5 Theoretical implications of Sparse Expansion

Recent theoretical work (Hänni et al., 2024; Adler & Shavit, 2024) investigates the algorithmic upper and lower bounds of polysemantic neuronal computation in toy examples. To explore empirical evidence along this body of work for real-world models, we investigate the improvements made by Sparse Expansion in Pythia-1.4B in 80% unstructured sparsity. We estimate the approximate number of effective features a set of inputs has by applying PCA to the set and finding the minimum number of components required to reach 90% explained variance. As expected, the average minimum required components for the inputs to the experts, weighted by the number of inputs in each group, decreases after clustering for every FFN weight matrix (Figure 8a).

To provide empirical evidence on the bounds of computation under entanglement, we explore modeling ability compared to the number of input features. To identify a bound for minimum error under sparse computation, we consider the RMSE of each clustered sparse output to the dense output, normalized to the overall RMSE for that layer as a proxy for computational ability. We compare this to the number of required PCA components for said cluster as before. Across all clusters in all layers of the network, there is a linear front that emerges in log-log scale: as the number of required components increases, so too does the minimum error (Figure 8b). Next, we consider the bound on maximum improvement in sparse computation under entanglement. When a cluster has fewer effective features, since each expert has the same number of parameters, Sparse Expansion allocates relatively more parameters to model these features than SparseGPT does, as the latter must account for all inputs. However, when a cluster has many required components, Sparse Expansion and SparseGPT allocate a similar amount of parameters, leading to relatively lower improvement. Therefore, performance improvements increase with fewer effective features. However, beyond a certain point, adding more features no longer yields further performance gains. This trend is also visible in the linear frontier of the log-log plot (Figure 8c). Thus, we provide some empirical demonstrations of the existence of bounds of both loss and improvement of sparse computation under entanglement.

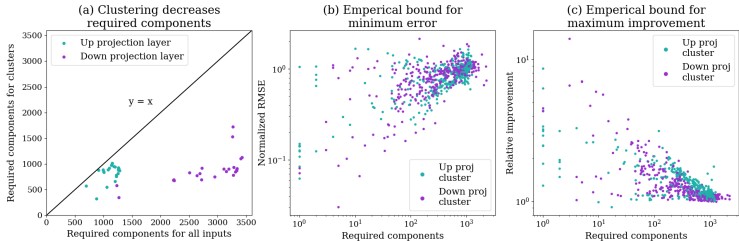

Figure 8: Empirical demonstrations of performance bounds. (a) As a result of clustering, the weighed average minimum number of components to capture 90% of the explained variance decreases for every layer. (b) As the number of required components for a particular cluster increases, so too must the error. (c) As the number of required components for a particular cluster decreases, Sparse Expansion improves more over SparseGPT, but up to a bound. Data collected in Pythia-1.4B.

### 3.6 SPARSE EXPANSION PERFORMANCE

We evaluate how well Sparse Expansion performs against other competitive one-shot pruning techniques, including in terms of inference speed (Table A3). Despite its leading evaluation performance (Figure 9; Table A1, A5), this method is likely not practically implementable without further optimizations to counteract the increase in memory footprint, including tuning the number of clusters per neuron (Figure A3). Nevertheless, we hope that Sparse Expansion serves as an inspiration for future sparsification techniques that address entanglement for better performance.

#### 3.6.1 MODELS, DATASETS, AND SETUP

We use the Pythia series of pre-trained LLMs to evaluate how Sparse Expansion performs across model sizes, from Pythia-70M to Pythia-12B. We further evaluate Sparse Expansion across the entire Llama-2 family. We use a subset of the Wikitext-2 train dataset as calibration data for input-aware pruning and evaluate using the corresponding test set through the perplexity metric. Furthermore, to evaluate the performance of Sparse Expansion in out-of-distribution (OOD) data, we evaluate the sparse model in 5 zero-shot standard benchmark tasks in both Llama and Pythia. For our performance benchmarks, we use 16 clusters at each level of routing in Sparse Expansion. We rely upon the RAPIDS library (Raschka et al., 2020) to accelerate the PCA and K-means models by orders of magnitude. We utilize and build upon the SparseGPT GitHub repository.

#### 3.6.2 PERFORMANCE ACROSS SCALES

We evaluate the performance of Sparse Expansion against other one-shot pruning techniques across a range of model sizes in Pythia and sparsities in Llama-2-7B (Figure 9). Across all model sizes of Pythia, Sparse Expansion outperforms all other pruning techniques at 50% unstructured sparsity, approaching dense performance as model size increases. Moreover, for Llama-2-7B, across all levels of sparsity, Sparse Expansion outperforms all other techniques. At higher levels of sparsity, the gap in performance between the techniques grows. We run further experiments on the entire Llama 2 family as well, and Sparse Expansion similarly outperforms other methods (Table A5). Finally, our experiments show Sparse Expansion outperforming contemporary pruning algorithms in OOD settings as well (Table A1, A2).

## 4 RELATED WORK

**Polysemanticity**   There is a plethora of ongoing research contributing to the understanding of polysemanticity in neural networks from a mechanistic interpretability perspective (Bricken et al., 2023; Huben et al., 2023; Lecomte et al., 2023; Templeton, 2024). These efforts primarily rely upon sparse autoencoders to disentangle output activations into human-interpretable features, losing information specific to individual neurons in the process. As we focus on neurons due to their direct role in network pruning, we derive our own formulation of entanglement as an extension of prior notions. There are also other works that investigate individual neuronal responses directly utilizing tech-

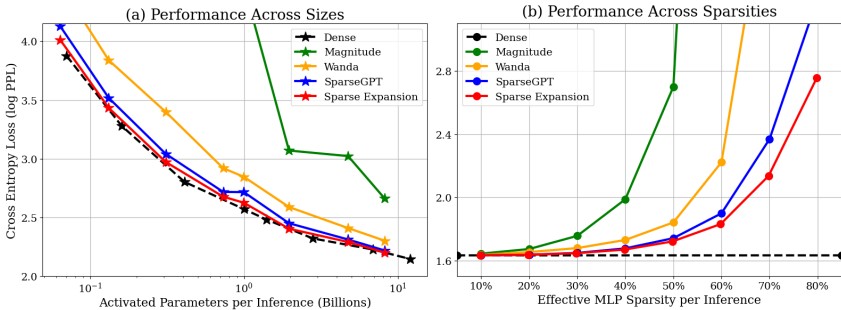

Figure 9: Sparse Expansion across model sizes and sparsities. (a) Performance comparisons on Wikitext-2 perplexity between Magnitude Pruning (MP), Wanda, SparseGPT, and Sparse Expansion on Pythia models from sizes of 70M parameters to 12B parameters. Every FFN in each model was sparsified to 50% sparsity. Each star represents a particular model size on the dense curve, and the corresponding sparsified model is the marker directly to its left on the sparse curves. (b) Performance for Llama-2-7B at different levels of sparsity for MP, Wanda, SparseGPT, and Sparse Expansion. The x-axis points in both graphs take into account the cost of routing.

niques such as sparse probing (Gurnee et al., 2023), as well as those that identify special neuron types in LLMs, such as Universal neurons (Gurnee et al., 2024) and Confidence Regulatory neurons (Stolfo et al., 2024). However, there is no recent literature tying polysemanticity and neuronal entanglement to sparse network performance.

**Compression**   A multitude of advanced weight pruning algorithms, such as Wanda (Sun et al., 2023) and SparseGPT (Frantar & Alistarh, 2022), and quantization algorithms (Kim et al., 2023; Dettmers et al., 2022; Ashkboos et al., 2024; Egiazarian et al., 2024; Dettmers et al., 2023; Zhao et al., 2023; Lin et al., 2024) exist. Most advanced algorithms are input-aware so as to specialize the weights to the most important input features. Other pruning approaches, such as SWAP (You & Cheng, 2024) and WD-based channel pruning (Duan & Li, 2020), have also used WD, though for the gradient of the loss or for channel similarity, rather than for analyzing neurons. While outliers in the features and weights are known to be the among the most challenging factors to address when quantizing to extremely low bits, no equivalent understanding has been made for high sparsities.

## 5    Conclusion and Discussion

In this work, we for the first time demonstrate the impact of neuronal entanglement on network performance under weight sparsity, a previously unexplored avenue. From our work and others, we suspect that every neuron is to some extent entangled, but that this entanglement of features is easier for some neurons to resolve than it is for others. We explore this notion of entanglement through our metric of mapping difficulty, and find that Wasserstein distance is a novel, highly pertinent indicator of entangled neurons that must differentiate similar inputs into different outputs. Furthermore, as Wasserstein neurons in particular are incredibly sensitive to sparsification, we posit that the robustness of a neuron to sparsity is directly dependent on its degree of entanglement. Finally, we have shown that our experimental framework Sparse Expansion is an effective way to disentangle the complex entangled state of a sparse neuron, and use it to explore computational bounds in empirical real-world models. The disentanglement provided by Sparse Expansion benefits Wasserstein neurons the most, providing further support that such neurons are the most entangled.

In future work, we plan to study Wasserstein neurons in the framework of mechanistic interpretability to understand what circuits they form. From our insight that more entangled neurons are harder to sparsify, we will investigate creating efficient, entanglement-aware sparsification algorithms to preserve performance at higher sparsities. Looking forward, perhaps just as outlier features and weights are well understood to be one of the most significant challenges when quantizing to fewer bits, so too can neuronal entanglement be understood as the challenge of pruning to higher sparsities.

## 6 ACKNOWLEDGMENTS

The authors would like to extend their gratitude to Lori Leu for her insightful comments on the application of the Wasserstein distance metric. We also wish to thank Elias Frantar for his help in working with the SparseGPT implementation and his advice for the project. Additionally, we would like to thank Tony Tong Wang and Thomas Athey for their valuable feedback and constructive discussions.

This work was supported by an NIH Brains CONNECTS U01 grant and AMD's AI & HPC Fund.

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

# A APPENDIX

## A.1 PSEUDO-CODE FOR SPARSE EXPANSION

Algorithm A1 describes the sparsification process of Sparse Expansion. The sparse experts are created in a layer-wise sequential fashion for each linear layer of every FFN transformer block to create the sparse model. Algorithm A2 refers to the inference procedure of Sparse Expansion once the model is pruned following the methods described in Algorithm A1 and Section 3.1.

---

**Algorithm A1** Sparse Expansion model generation. The following layerwise procedure can be repeated for each linear layer in the transformer.

---

1: **procedure** LAYERWISE SPARSE EXPANSION SPARSIFICATION PROCESS
2:     $\{x\} \leftarrow x_i \in \mathbb{R}^n$   //set of calibration inputs to layer
3:     $W \leftarrow m \times n$   //layer weights
4:     $c$   //number of clusters
5:     $r$   //factor to reduce dimensionality by
6:     $R \leftarrow \mathbf{PCA}(\frac{n}{r})$   //new PCA object with $\frac{n}{r}$ components
7:     $R.\mathbf{fit}(\{x\})$   //fit $R$ to inputs
8:     $K \leftarrow \mathbf{Kmeans}(c)$   //new K-means object with $c$ initial centroids
9:     $K.\mathbf{fit}(\{R(x)\})$   //fit $K$ to dimensionality reduced inputs
10:    **for** $j = 1, 2, 3...c$ **do**
11:       $X_j \leftarrow \{x | K(R(x)) = j\}$   //group $\{x\}$ into its component clusters
12:       $W_j \leftarrow W$   //make a copy of the original weight matrix
13:       $S_j \leftarrow \mathbf{SparseGPT}$   //make a SparseGPT object
14:       $W_j' \leftarrow S_j.\mathbf{sparsify}(W_j, X_j)$   //sparsify $W_j$ using $X_j$

---

**Algorithm A2** Sparse Expansion inference. The following layerwise procedure is repeated at inference time for each clustered layer.

---

1: **procedure** LAYERWISE SPARSE EXPANSION INFERENCE PROCESS
2:     $\{x\} \leftarrow x_i \in \mathbb{R}^n$   //set of inputs to layer
3:     $\{W\}$   //set of experts
4:     $R$   //PCA model
5:     $K$   //K-means model
6:    **for** $i = 1, 2, 3...$ **do**
7:       $j \leftarrow K(R(x_i))$   //find the cluster assignment of $x$
8:       $y_j \leftarrow W_j(x_i)$   //run inference with the correct expert

---

## A.2 DISTRIBUTION OF WASSERSTEIN DISTANCES ACROSS ALL LLAMA-2-7B FFN LAYERS

After collecting the Wasserstein distance to the normal distribution for every neuron, we find that all up and gate projection matrices in each Llama-2-7B FFN block have high WD neurons. We also find that certain down projection matrices also have high WD neurons, though most do not.

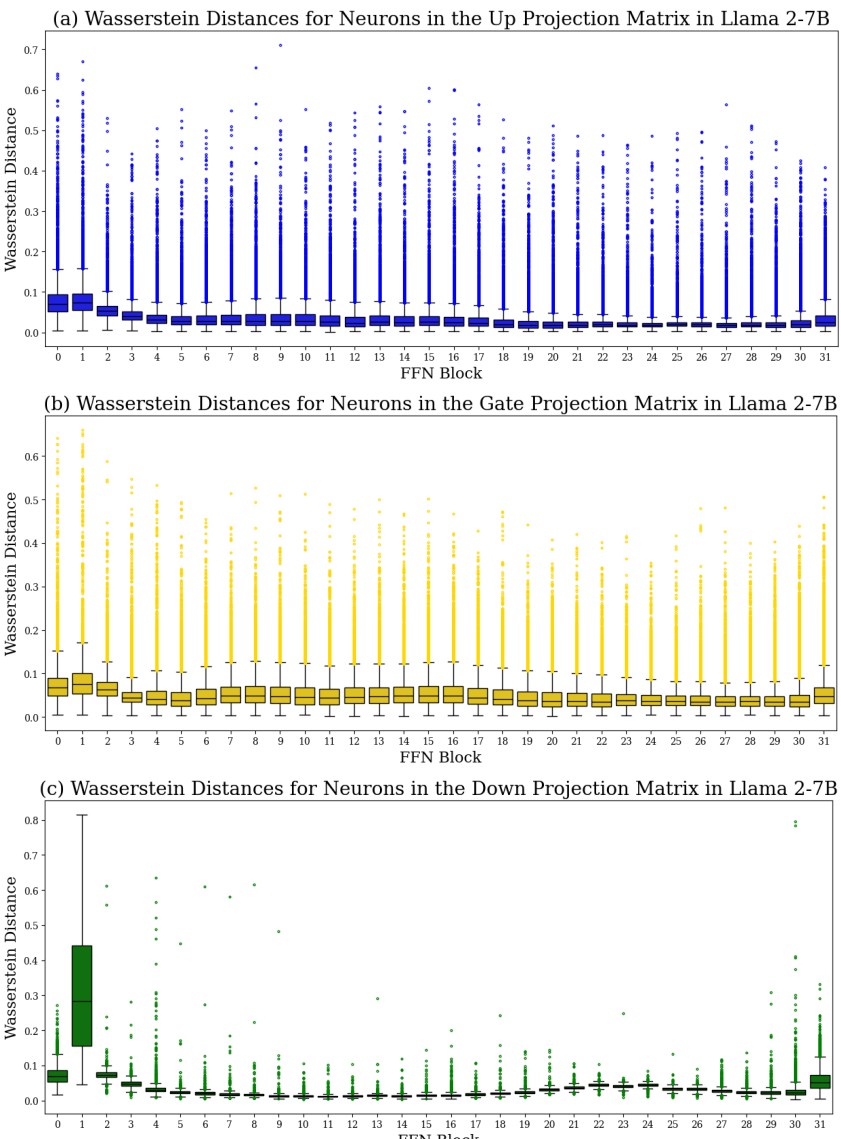

Figure A1: High Wasserstein distance neurons in each layer. Many neurons with a high WD to the Gaussian distribution exist in every FFN block, and in every up (a) and gate projection (b) specifically. Certain down projection layers also have high WD neurons (c). The box plots show the range of non-outliers, as well as the first quartile, the median, and the third quartile of neuronal WD. The outliers are defined as 1.5 times the interquartile range less than the first or more than the third quartile and are represented by the points.

## A.3 SPARSE EXPANSION PERFORMANCE IN OUT-OF-DISTRIBUTION DATA

We evaluate the performance of Llama-3.2-1B[2] (Table A1) and Pythia-1.4B (Table A2) on a range of natural language modeling tasks, including ARC-e (Easy) and ARC-c (Challenge) for arithmetic reasoning, Lambada (Paperno et al., 2016) for contextual word prediction, SciQ (Welbl et al., 2017) for scientific question answering, and MMLU for multitask general knowledge assessment. As dense Pythia-1.4B does not score better than random chance on MMLU, we do not benchmark it on this task. We compare various pruning algorithms at 50% sparsity, including Magnitude Pruning (MP), Wanda, SparseGPT, and Sparse Expansion with 16 clusters, to the dense baseline. Sparse Expansion consistently excels across both models, achieving the highest scores on tasks among sparsification algorithms.

Table A1: Performance of Llama-3.2-1B under different pruning algorithms.

| Algorithm | Sparsity | ARC-e | ARC-c | Lambada | SciQ | MMLU |
|---|---|---|---|---|---|---|
| Dense | 0% | 65.488 | 31.314 | 53.969 | 91.4 | 37.701 |
| Magnitude | 50% | 45.244 | 22.354 | 4.677 | 67.1 | 23.493 |
| Wanda | 50% | 50.800 | 23.635 | 31.457 | 85.2 | 25.428 |
| SparseGPT | 50% | 55.640 | 24.403 | 31.613 | 86.8 | 25.046 |
| **Sparse Expansion** | 50% | **57.713** | **26.962** | **35.807** | **87.5** | **28.729** |

Table A2: Performance of Pythia-1.4B under different pruning algorithms.

| Algorithm | Sparsity | ARC-e | ARC-c | Lambada | SciQ |
|---|---|---|---|---|---|
| Dense | 0% | 61.742 | 27.389 | 48.981 | 86.9 |
| Magnitude | 50% | 42.003 | 19.198 | 1.533 | 69.0 |
| Wanda | 50% | 54.630 | 23.976 | 45.041 | 85.7 |
| SparseGPT | 50% | 56.608 | 24.061 | 44.615 | 85.8 |
| **Sparse Expansion** | 50% | **58.449** | **25.720** | **46.424** | **86.3** |

---

[2]https://ai.meta.com/blog/llama-3-2-connect-2024-vision-edge-mobile-devices/

## A.4    NEURONAL ENTANGLEMENT TRAJECTORY ACROSS TRAINING IN LLMS

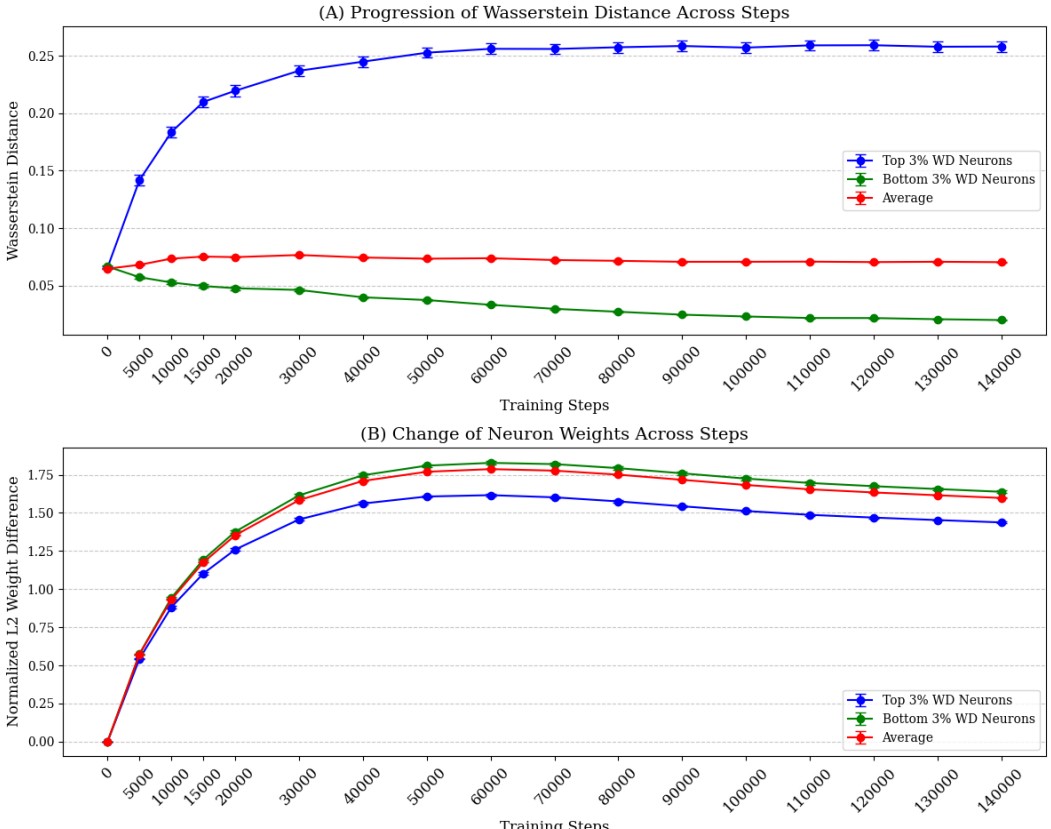

Figure A2: Analyzing neuronal entanglement during training. The intermediate checkpoints of Pythia-1.4B are available over the course of its training, from initialization to completion. Thus, we collect data from 17 different checkpoints over the course of its training, first at intervals of 5,000 steps, then at intervals of 10,000 steps after step 20,000. (a) We calculate each neuron's output distribution WD to a Gaussian as before in Equation 1. We do so for each training step. From the WD of neurons in the last training step, we separate out the top 3% of neurons with the highest WD and the bottom 3% of neurons with the lowest WD. We also find the average WD across all neurons. The progression of neuronal WD across training reveals that all neurons initially exhibit a Gaussian-like distribution, as expected, but some neurons rapidly differentiate into entangled neurons with very high WD and within just 5,000 steps (corresponding to approximately 10 billion tokens). The WD of such neurons then levels off afterward. (b) Using the same groups as in (a), we visualize the change in neuronal weights. We calculate the $L^2$ norm between each neuron's weights at each training step and its weights at model initialization (step 0), and normalize this value by the $L^2$ norm of the neuron's weights at initialization. Notably, neurons with high WD do not demonstrate more changes in their weights over the course of training than the average neuron, or neurons with low WD. Error bars represent one standard error of the mean. Neurons are from the up projection matrix in the second FFN block of Pythia-1.4B.

A.5 Optimizations for practical implementation

To evaluate the inference latency of Sparse Expansion we implemented a Sparse Expansion layer based on PyTorch and optimized sparse-quantized GPU kernels called Sparse Marlin (Frantar & Alistarh, 2024; Castro & Alistarh, 2024), which supports the INT4 + 2:4 sparsity format. To better utilize the compression kernel, we use both sparsification and quantization to demonstrate speedup. We use a linear layer of appropriate size as an upper bound approximation for our router cost, which is followed by 4 bit, 2:4 sparse matrix multiplication. We have run the layer-wise benchmarks for the typical layers sizes from Llama models on a single RTX3090 GPU. We can see in Table A3 that Sparse Expansion allows us to get up to a 4.8× speedup over the dense baseline. The speedup comes from the highly-compressed linear layer representation. Although there is overhead compared to a regular compressed matrix due to the presence of the router, such overhead decreases as layer size increases.

Table A3: Sparse Expansion inference speedup. Layer-wise single batch inference latency (in $\mu$s). The layer sizes are chosen specifically to match the layers of Llama-2-7B and Llama 2 70B.

| Layer Size | 4k × 12k | 4k × 22k | 11k × 4k | 8k × 10k | 8k × 57k | 28k × 8k |
|---|---|---|---|---|---|---|
| Dense | 132 | 227 | 114 | 220 | 1168 | 556 |
| Sparse Expansion | 76 | 76 | 75 | 76 | 241 | 138 |
| Speedup | 1.7× | 3.0× | 1.5× | 2.9× | 4.8× | 4.0× |
| Sparse | 26.8 | 44.7 | 24.4 | 42.3 | 216 | 109 |
| Overhead | 2.9× | 1.7× | 3.1× | 1.8× | 1.1× | 1.3× |

Additionally, we investigate how many experts different neurons need to improve performance. We find the relative improvement of each neuron, as defined in Section 3.3, across a different number of total experts. Specifically, we choose 2, 4, 8, and 16 experts for Sparse Expansion, compared to SparseGPT with its single expert. In this setting, we analyze the top 3% of neurons with the highest WD as well as the bottom 3% of neurons with the lowest WD, as defined before (Equation 1). We observe that Wasserstein neurons benefit far from Sparse Expansion than average for increasing clusters (left). Additionally, we split neurons into decile groups based on their relative improvement at 16 clusters. We find that, indeed, certain groups of neurons benefit very little from further additional experts past eight experts (right). Thus, further optimizations can be made to reach a balance between performance and memory increase.

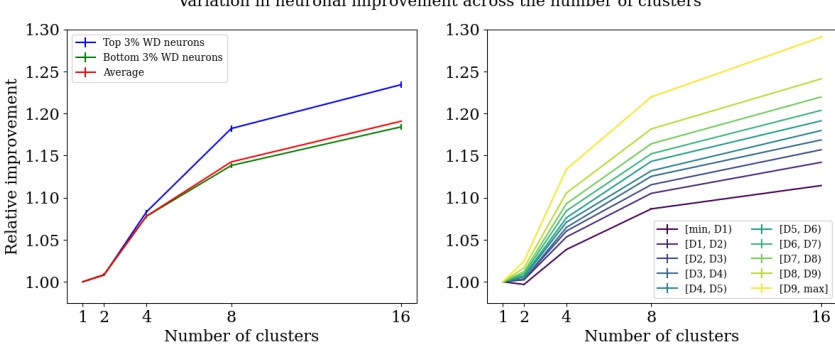

Figure A3: Improvement across clusters for different groups of neurons. (a) Wasserstein neurons benefit much more from Sparse Expansion than average with increasing clusters. (b) Different deciles of neurons have varying degrees of improvement from Sparse Expansion. D$n$ indicate the deciles from D1 to D9. The decile groups are decided by their relative improvement at 16 clusters. For example, the first decile group consists of relative improvements between the minimum and D1 at 16 clusters, the second decile group consists of relative improvements between D1 and D2 at 16 clusters, and so on. Error bars represent one standard error of the mean. Neurons from the up projection matrix of the second FFN block of Pythia-1.4B.

### A.6 Wasserstein neurons do not have particularly high weights

To understand whether Wasserstein neurons arise from having substantially higher weights than average, we measure the mean weight magnitude for each neuron. We find that Wasserstein neurons do not have weight magnitudes that are particularly above average; if anything, there seems to be a slight negative correlation between a neuron's WD and its mean weight magnitude (Figure A4a). To investigate how this difference affects sparsification, we sparsify this layer to 80% unstructured sparsity via SparseGPT, calibrated with Wikitext-2. This takes into account both weight magnitude as well as the influence of the inputs via the Hessian matrix. Wasserstein neurons are especially sensitive to sparsity and have an outsized impact on model performance, even more so than neurons with high average weight magnitude (Figure 3). However, these neurons are sparsified slightly more than average by this current advanced sparsification approach. This suggests that future sparsification schemes should take into account a neuron's WD and degree of entanglement, rather than just its weights and the Hessian.

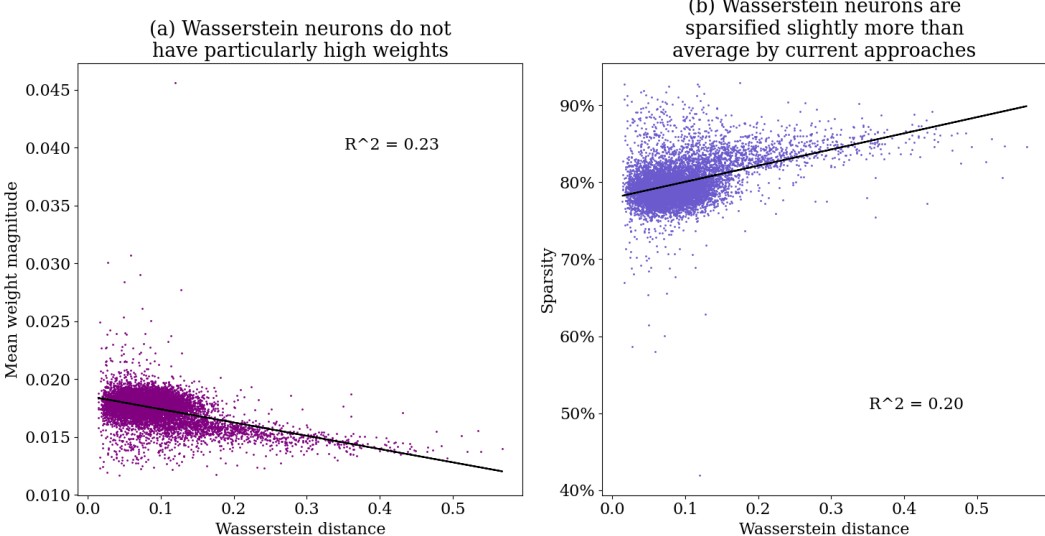

Figure A4: Wasserstein neurons do not have particularly large weights, in terms of their average magnitude, and so are sparsified slightly more. (a) Neurons with high WD do not have large average weight magnitudes. Of the top 3% of neurons with the highest WD, just one is also within the top 3% of neurons with the largest average weight magnitudes. (b) Partially as a result of their lower than average weights, Wasserstein neurons tend to be sparsified slightly more than average in an unstructured setting. The top 3% Wasserestein neurons are sparsified 6% more than average. The neurons are from the up projection of the second FFN in Pythia-1.4B.

### A.7 DIMINISHING RETURNS FOR KEEPING ENTANGLED NEURONS FULLY DENSE

As shown in Figure 3, entangled neurons are particularly sensitive to pruning. We design an experiment to understand the opposite effect on model performance, namely of keeping the Wasserstein neurons dense. In our setup, we selectively keep the top $x\%$ of Wasserstein neurons dense, while pruning each of the remaining neurons to a sparsity of $\frac{s}{100-x}\%$ to maintain an overall target sparsity of $s\%$. This approach is compared against a baseline where all neurons are pruned to the same sparsity percentage $s\%$, abbreviated as same sparsity per neuron (SSPN) in the table and equivalent to $x = 0\%$.

We conduct this experiment on Llama-2-7B. We use SparseGPT to sparsify the neurons, use part of the Wikitext-2 train set as calibration data, and evaluate on the Wikitext-2 test set. As illustrated in Table A4, keeping Wasserstein neurons dense at the cost of sparsifying every other neuron more to achieve a target sparsity does not enhance model performance. Additionally, model performance worsens progressively as the proportion of neurons kept dense ($x\%$) increases, since now less entangled neurons are also being kept dense. This behavior is likely due to the fact that the benefit of allowing Wasserstein neurons to retain all of their weights is outweighed by the cost of sparsifying every other already sparse neuron even more.

Table A4: Perplexity of Llama-2-7B on Wikitext-2 while sparsifying to $s\%$ overall and preserving $x\%$ of Wasserstein neurons.

|  | $s = 50\%$ | $s = 60\%$ | $s = 70\%$ | $s = 80\%$ | $s = 90\%$ |
|---|---|---|---|---|---|
| **SSPN ($x = 0\%$)** | **6.219** | **7.420** | **12.73** | **33.26** | **366.0** |
| $x = 3\%$ | 6.259 | 8.023 | 14.70 | 40.40 | 395.7 |
| $x = 5\%$ | 6.345 | 8.131 | 16.03 | 46.67 | 629.5 |
| $x = 7\%$ | 6.366 | 8.547 | 17.37 | 61.95 | 978.3 |
| $x = 10\%$ | 6.522 | 9.232 | 19.48 | 79.53 | 8066 |

## A.8 DERIVING MAPPING DIFFICULTY AS A METRIC OF ENTANGLEMENT

We show more reasoning behind the choice of the normalizing factors $N_{\boldsymbol{x}}$ and $N_y$ in Equation 2. First, we choose $N_{\boldsymbol{x}} = \max_{1 \leq i < j \leq n}\{||\boldsymbol{x}_i - \boldsymbol{x}_j||\}$ to be the maximum $L^2$ norm between a pair of inputs to simply scale all $L^2$ norms to be between 0 and 1. Next, we choose $N_y = \mathrm{median}_{1 \leq i < j \leq n}\{||y_i - y_j||\}$ to be the median based on the following observations. Specifically, we would like to preserve and highlight the fact that there are many IO pairs that have a relatively low difference in their inputs, but are mapped to very different outputs, one group of which is circled in purple in Figure A5d.

First, we considered using the maximum $L^2$, as we did for $N_{\boldsymbol{x}}$. However, note that there is a very small number of samples that drives the maximum to be much further away from meaningful data points in both the random and Wasserstein neurons. Next, we considered the mean. Due to the presence of the outlier data points of interest that have a much greater difference in output than expected, the mean is also driven much higher for the Wasserstein neuron. Indeed, observe that for the Wasserstein neuron, the mean is much greater than the mode, while they are much more similar for the random neuron. We therefore use the median to normalize for inter-neuron differences in the expected range of their output differences, but to also be robust to outliers that would obfuscate the IO pairs of interest through an inflated mean.

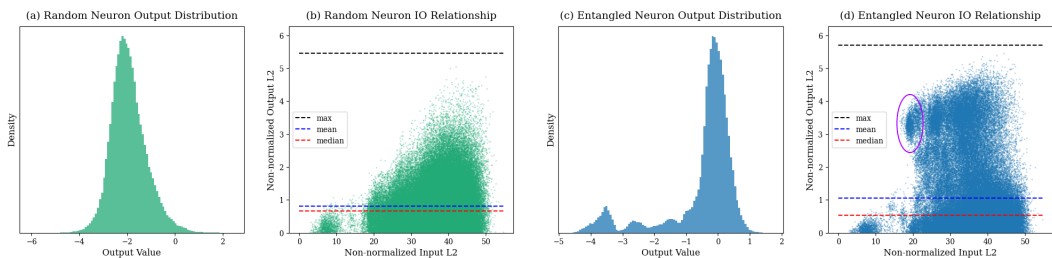

Figure A5: Deriving a normalization constant for the difference in outputs. (a) The output distribution of a random neuron. (b) The non-normalized relationship between the $L^2$ norm between pairs of inputs and the $L^2$ norm between their corresponding outputs for the random neuron. (c) The output distribution of a Wasserstein neuron. (d). The non-normalized relationship between the $L^2$ norm between pairs of inputs and the $L^2$ norm between their corresponding outputs for the Wasserstein neuron. Note how the mean is much higher than the median. One group that has a much higher output $L^2$ norm than expected for its relatively low input $L^2$ norm is circled in purple. These are the same neurons from Figure 2.

## A.9 ADDITIONAL CLUSTERS IMPART MORE PERFORMANCE

To understand how Sparse Expansion scales with the number of experts per linear layer, we test its performance from 2 to 32 experts. Interestingly, with 2 experts, very little performance benefits are realized. However, with each doubling of experts following 2 experts, we realize a nearly constant linear improvement in perplexity.

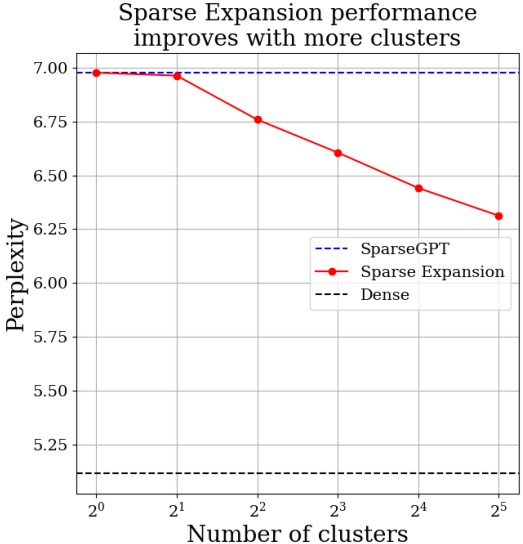

Figure A6: Increasing the number of clusters improves Sparse Expansion performance in Llama-2-7B.

### A.10 FURTHER EXAMPLES OF DISENTANGLEMENT

We present additional evidence of neuronal disentanglement in Pythia-1.4B and Llama-2-7B. Figure A7 shows the recovery of the dense output distribution with increasing experts in Llama-2-7B. This is analogous to Figure A9, where we see a similar trend in Pythia models. Clustering gradually decreases the WD of the sparse output to that of the dense, thus improving upon SparseGPT, equivalent to the single cluster case. Moreover, this results in direct improvement of model performance as depicted in figure A6.

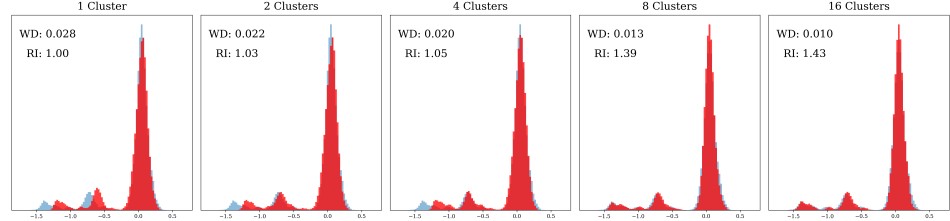

Figure A7: Modeling recovery with more experts in Llama-2-7B. Use of more experts can recover the dense output distribution even at very high sparsity, which is set to 90% for each expert. This is the same neuron from Figure 1d.

Analogous to Figure 1, we observe the effect of clustering inputs on a random neuron and an entangled neuron in the gate projection of the second FFN of Pythia-1.4B. SparseGPT fails to capture the output distribution of the high WD neuron as it does for a random neuron. With clustering via Sparse Expansion, both neurons improve, but the entangled neuron improves more. The granular analysis of the component clusters within both neurons reveals the specialization to vastly different parts of the output distribution in the entangled neuron as compared to the normal neuron (Figure A8).

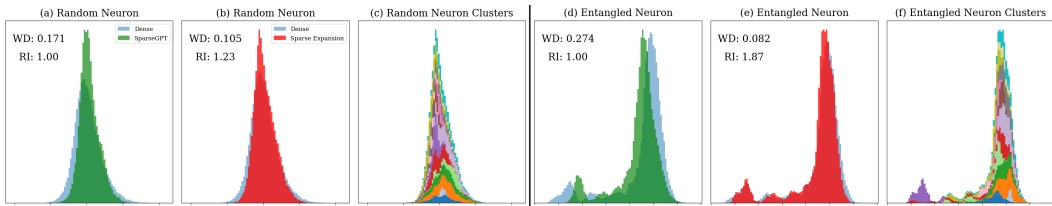

Figure A8: Sparse Expansion disentangles neurons in Pythia-1.4B. The dense output distribution of a random neuron, along with its sparse via SparseGPT (a) and via Sparse Expansion (b) sparse output distributions. The dense output distribution of a random neuron, along with its sparse via SparseGPT (d) and via Sparse Expansion (e) sparse output distributions. For both the random and entangled neuron, component clusters are shown in a distinct color to visualize their range (c, e). WD represents the Wasserstein distance between the Sparse Expansion sparse output distribution and the dense distribution. RI represents relative improvement. These are the same neurons from Figure 2.

## A.11 PERFORMANCE ACROSS THE LLAMA FAMILY

We analyze the performance of Sparse Expansion against other sparsification algorithms across all members of the Llama-2 family—Llama-2-7B, Llama-2-13B, and Llama-2-70B—both under sparsity and joint sparsity-quantization compression (Touvron et al., 2023).

Table A5: Sparse Expansion across the Llama family.

|  | Sparsity | Bits | Llama-2-7B | Llama-2-13B | Llama-2-70B |
|---|---|---|---|---|---|
| Dense | 0% | 16-bit | 5.1168 | 4.5736 | 3.3192 |
| MP | 50% | 16-bit | 16.029 | 6.8270 | 4.9846 |
| Wanda | 50% | 16-bit | 6.7757 | 5.8527 | 4.0219 |
| SparseGPT | 50% | 16-bit | 5.7082 | 5.0521 | 3.9013 |
| **Sparse Expansion** | 50% | 16-bit | **5.5839** | **4.9728** | **3.8791** |
| SparseGPT | 2:4 | 16-bit | 6.9767 | 5.9934 | 4.8002 |
| **Sparse Expansion** | 2:4 | 16-bit | **6.4456** | **5.6255** | **4.6671** |
| SparseGPT | 2:4 | 4-bit | 7.2759 | 6.1101 | 4.9036 |
| **Sparse Expansion** | 2:4 | 4-bit | **6.5745** | **5.7151** | **4.7586** |
| SparseGPT | 2:4 | 3-bit | 13.076 | 6.5055 | 5.2552 |
| **Sparse Expansion** | 2:4 | 3-bit | **7.0757** | **5.9872** | **5.0588** |

Sparse Expansion outperforms all other pruning techniques for both 50% unstructured sparsity as well 2:4 sparsity in all Llama models (Figure A5). In addition to non-quantized sparsity, we consider how Sparse Expansion performs in the context of compression with 2:4 structured sparsity and quantization via GPTQ (Frantar et al., 2022). We first sparsify each linear layer in each FFN block to 2:4 sparsity, then quantized to 3 and 4 bits. Our method outperforms SparseGPT across all models and across both conditions and in all models (Figure A5).

Across multiple model sizes, sparsity and compression levels, and advanced models, Sparse Expansion attains state-of-the-art performance for post-training one-shot sparsification when compared to other highly competitive pruning techniques. We do so by leveraging the powerful pruning algorithm of SparseGPT and combining it with input specialization to utilize the insights we gain from how entangled neurons behave under sparsity.

Because GPTQ (Frantar et al., 2022), a leading post-training quanization scheme, also relies upon the Hessian matrix for its algorithm, we combine it with SparseGPT for combined one-shot compression. Sparse Expansion also outperforms native SparseGPT and GPTQ across all compression settings.

## A.12 EFFECT OF SPARSITY ON NEURONAL OUTPUT DISTRIBUTIONS

With increasing sparsity, the sparse output distributions of the high WD neurons and random neurons converge toward the normal distribution (Figure A10). A specific example of a neuron is shown in Figure A9.

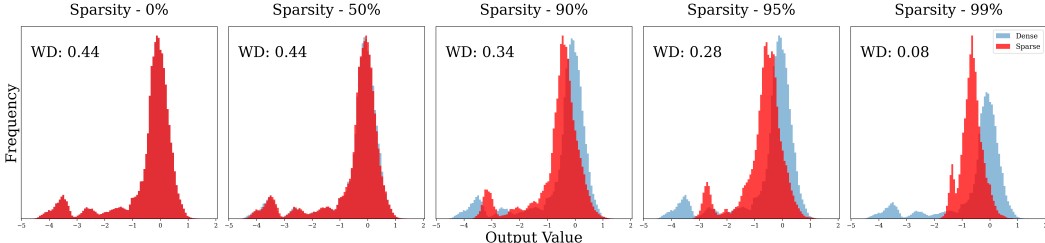

Figure A9: Increasing sparsity induces normality. A highly entangled neuron's dense distribution (blue) and sparse distribution (red). As sparsity increases, the output distribution of the sparse neuron becomes progressively more Gaussian. WD represents the Wasserstein distance. This is the same neuron from Figure 2c.

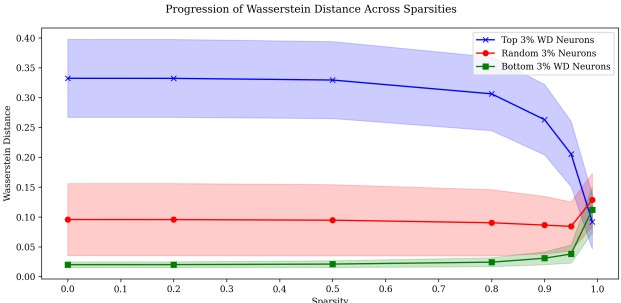

Figure A10: Output distributions become more normal under sparsity. The Wasserstein distance between a neuron's normalized sparse output distribution and the Gaussian distribution is shown as sparsity increases for the top 3% of entangled Wasserstein neurons, the same number of bottom 3% WD neurons, and a random sample of 3% of the neurons. For highly entangled neurons, the WD decreases significantly at higher sparsities whereas it remains more or less constant for the bottom 3% of neurons and for the random neurons. Range indicates maximum and minimum WD for a group. Data collected from of the second up projection matrix in Pythia-1.4B.

Furthermore, with increasing sparsity, the magnitudes of the means and variances of each neurons' sparse output distribution both shift toward zero. This is reasonable, as with fewer nonzero weights to combine together features, both the mean and variance should decrease in magnitude.

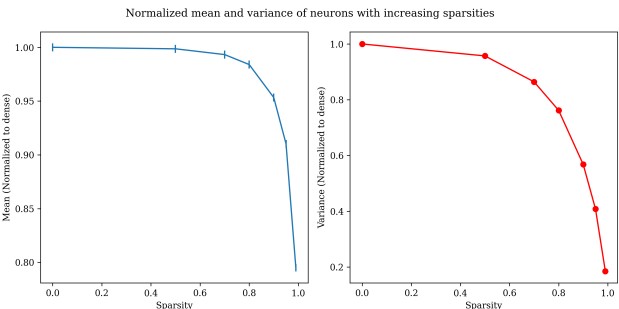

Figure A11: Mean and variance shift toward zero under sparsity. Across all neurons, with increasing sparsity, the magnitude of the mean of output distribution (left) and the variance of the output distribution (right) both tend toward zero. Both mean and variance have been normalized to their dense values. Error bars represent one standard error. Data collected from of the second up projection matrix in Pythia-1.4B.

### A.13 ALL NEURONS IMPROVE, BUT ENTANGLED NEURONS IMPROVE MORE AT HIGHER SPARSITIES

Measuring the relative improvement of each neuron through Sparse Expansion, we find that all neurons improve as a result of Sparse Expansion across both Pythia-1.4B and Llama-2-7B. Thus, we believe that every neuron has some level of innate entanglement, and so all neurons can be and are improved. Interestingly, we note that, with increasing sparsity, highly entangled Wasserstein neurons tend to improve more.

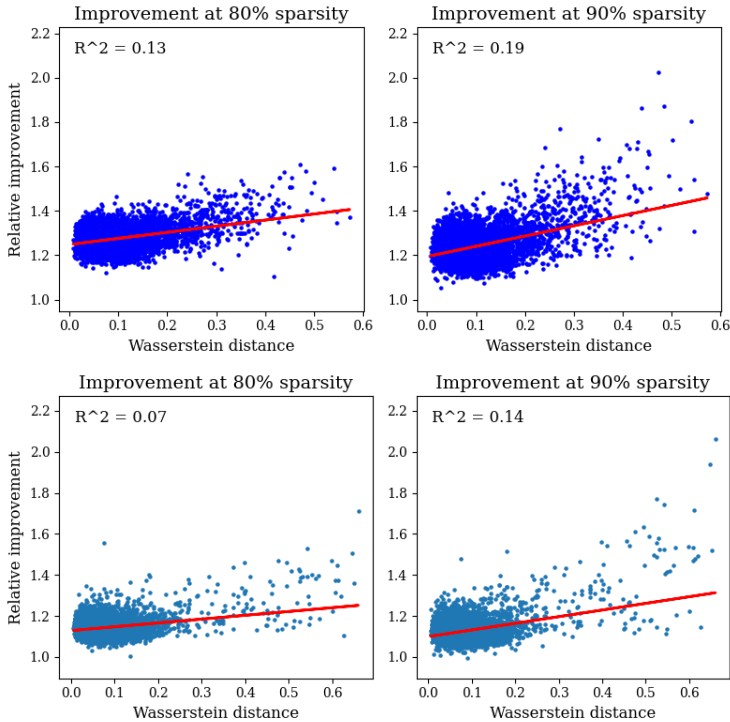

Figure A12: Entangled neurons improve more at higher sparsities. Relative improvement of each neuron in the second up projection matrix in Pythia-1.4B (top row) and in the second gate projection matrix in Llama-2-7B (bottom row) with respect to their WD from the Gaussian. Two sparsity levels, 80% and 90%, are shown. Sparse Expansion improves the expressibility of every neuron, thus improving performance. However, the entangled neurons improve more with higher sparsities, as visible in the right column.

## A.14 THE WASSERSTEIN DISTANCE BEST CAPTURES WHICH NEURONS IMPROVE

We also consider whether the magnitude of the mean of the output distribution or the variance of the distribution would be good predictors of the degree of neuronal improvement through Sparse Expansion. However, across both Pythia-1.4B (Figure 7) and Llama-2-7B (Figure A13), the Wasserstein distance from the normal is a better predictor of relative improvement, as defined previously. Though there is some correlation of the magnitude of the mean and variance of the output distribution with the relative improvement in Pythia-1.4B, that is not the case in Llama-2-7B. Furthermore, using the WD to predict neuronal improvement yields the highest coefficient of determination, $R^2$, across both models.

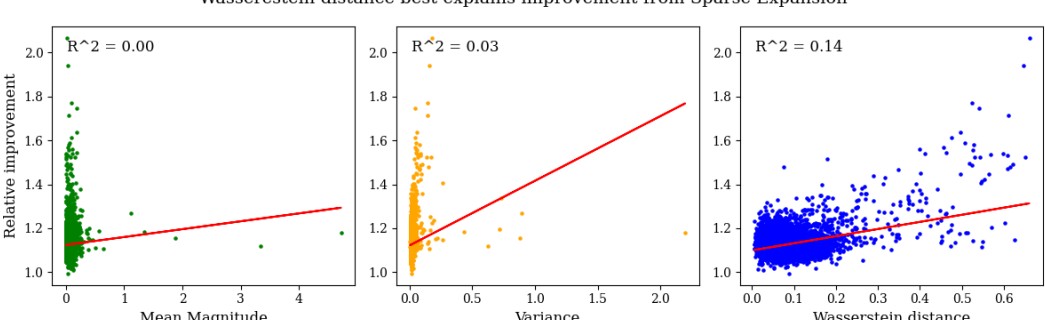

Figure A13: Wasserstein distance best captures improvement. Relative improvement of each neuron in the second gate projection matrix in Llama-2-7B with respect to the magnitude of the mean, variance, and Wasserstein distance from normal of the dense output distribution. Neurons pruned to 90% sparsity.

## A.15 OUTPUT DISTRIBUTIONS OF ENTANGLED NEURONS IN PYTHIA AND LLAMA

Figures A14 and A15 show the non-trivial, non-Gaussian output distribution of a subset of neurons from the Pythia-1.4B and Llama-2-7B models, illustrating examples of entangled neurons. We observe such neurons in every FFN block of the LLMs we investigated and believe that the existence of these neurons is a global phenomenon in transformers.

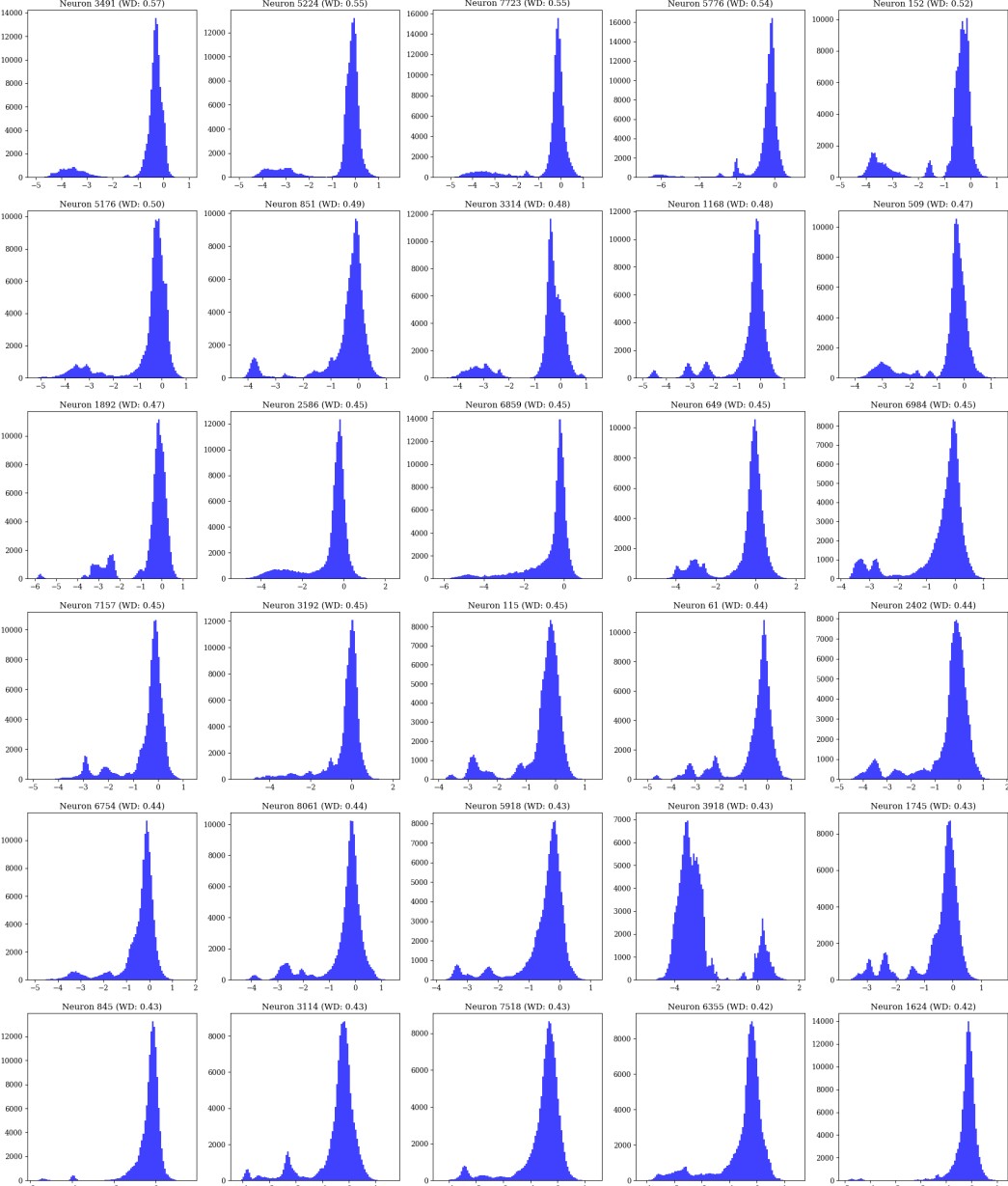

Figure A14: Dense output distributions of top 30 high WD neurons in Pythia-1.4B. The distributions are shown for the neurons of the up projection matrix in the second FFN block.

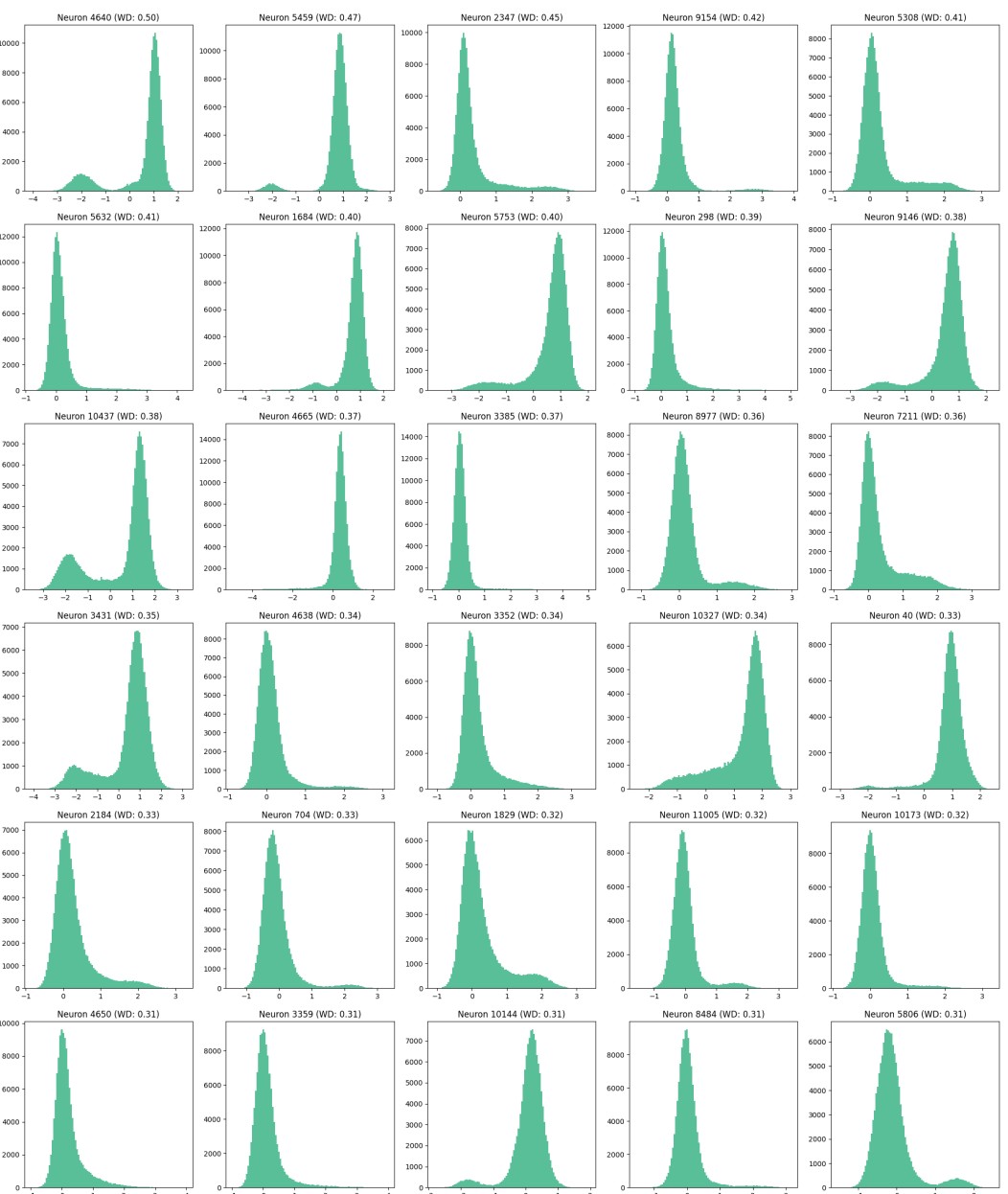

Figure A15: Dense output distributions of top 30 high WD neurons in Llama-2-7B. The distributions are shown for the neurons of the up projection matrix in the sixteenth FFN block.

