# OpenReview forum: "Wasserstein Distances, Neuronal Entanglement, and Sparsity"
_ICLR.cc/2025/Conference — ICLR 2025 Spotlight_

### Official Review · Reviewer_2v2T · 2024-10-16

**Soundness:** 4
**Presentation:** 2
**Contribution:** 3
**Rating:** 8
**Confidence:** 4

**Summary:**

In this work, the relevance of individual neurons in LLMs is estimated by comparing their normalized output distribution before the activation function to a Gaussian using the Wasserstein distance measure.
They find that the output of a small group of neurons deviates significantly from the gaussian distribution, and that these neurons contribute significantly to the model's performance. They define that a neuron is particularly entangled when it has to produce different outputs for similar inputs. They find that neurons that deviate from a gaussian distribution are particularly entangled.
They provide a framework that disentangles the neurons to some extent, that is, they reduce the Wasserstein distance measure. Using this framework, they obtain a lower loss on Wikitext-2 compared to other techniques when creating a sparsified model.

**Strengths:**

1) The relevance of individual neurons is studied in detail, which is interesting from a technical point of view when reducing the model size, but also points to areas that explainable AI researchers should look for.

2) A potential application in terms of model reduction is provided.

**Weaknesses:**

1) The exact sparsification algorithm used to generate Figure 2 is not given in the manuscript.

2) There are some problems with the Figures: Figures 1, 3, and 7 lack x and y labels.
In Figure 10, the caption does not match what is shown in the figure.

**Questions:**

1) The way I understand your sparsification algorithm in Figure 2 is this: For 3% of the neurons, you set 95% of the weights going into them to zero. Is this correct? And if so, how do you choose these weights?

2) Figure 10: Which model is used? The caption speaks of many, but the figure shows only one, as far as I understand it.

Line 422 typo: "the the"
Line 428 typo: "as" -> has

---

> ### Author Response · Authors · 2024-11-21
> **Official Rebuttal by Authors**
>
> Thank you very much for your review! We provide individual responses to your concerns below:
>
> 1. We thank the reviewer for pointing out this oversight. We should have included the exact sparsification procedure in Figure 3 (previously Figure 2), and this has now been detailed in the updated draft. Specifically, we use the SparseGPT sparsification algorithm to sparsify each targeted neuron to the same degree. The reviewer’s understanding of the scheme is correct - we target 3% of the neurons (which have either high WD, high mean, etc.) and use the SparseGPT algorithm to sparsify each of these neurons to the designated sparsity. SparseGPT selects weights and performs weight updates using a second-order approximation of the loss with respect to the parameters within each layer. Thus, this experiment retains essentially the most crucial weights for particular groups of neurons and prunes away the rest.
> 2. Regarding Figures 1, A8 and 6 (previously Figures 1, 3, and 7), the x axis label should be the value of the neuronal output and the y axis label should be frequency, and we have updated the figures to reflect this.
>
>     For Figure 9a (previously Figure 10a), we actually use Pythia 70M to Pythia 12B (70M, 160M, 410M, 1B, 1.4B, 6.9B, and 12B). We see now how the x-axis label of “Activated Parameters per Inference” can be confusing. To explain this, each point on a sparsification method curve represents the total activated parameters in an inference pass for a single one of these models when it is sparsified to 50% sparsity. This size also accounts for the slight increase due to the clustering inference for Sparse Expansion. For the line indicating dense performance, we have now included points along the curve to show the original model sizes, and have changed the markers to distinguish it from Figure 9b. Now, each sparsified model marker on the sparsification curves corresponds to the dense model directly to the right of it on the dense model curve.  For example, the first marker in the Sparse Expansion curve is the spasified version of the first marker in the dense model curve, Pythia 70M, the latter of which is directly to the right of the former. For Figure 9b, data from the entire graph is collected on Llama 2 7B with its MLP layers sparsified to a different degree, also with the clustering inference pass included in the calculation. We have updated the figure and the caption to improve clarity.
> 3. We have resolved these typos and also thoroughly revised the paper for readability and other typos as well.
>
> Thank you once again for your time in reviewing our paper, we really appreciate the feedback and we hope to have addressed your concerns!

---

> > ### Author Response · Authors · 2024-11-24
> > **Follow Up**
> >
> > We sincerely appreciate the reviewer’s time and effort in reviewing the paper. The reviewer’s insightful suggestions about the clarity of some key figures have helped improve the paper from its initial version. If the reviewer has any further questions or additional feedback, we would be more than happy to address them. Thanks!

---

### Official Review · Reviewer_uAy8 · 2024-11-02

**Soundness:** 2
**Presentation:** 1
**Contribution:** 2
**Rating:** 6
**Confidence:** 3

**Summary:**

The authors propose to compute the distance between the output distribution of a neuron in a neural network and a gaussian using Wasserstein distance as a measure of entanglement, i.e. a neuron could be highly polysemantic and thus entangled. It’s shown that some of the neurons in LLMs are highly entangled under the Wasserstein distance metric and sparsifying them leads to significant performance degradation compared to sparsifying random neurons. The authors propose sparse expansion by separating the input vectors and creating a mixture of neurons with low Wasserstein distance. The proposed sparse expansion methods outperform all other baselines under sparsity / pruning.

**Strengths:**

The proposed sparse expansion seems to outperform SparseGPT and all other baselines, though I’m not in this subfield and I don’t know what are the strong and weak baselines here.

**Weaknesses:**

- It’s very unclear what do you mean by the gaussian output distribution of a single neuron. For example, the introduction section says that a column of a weight matrix is a neuron. Here it’s unclear what’s the matmul operation between input and weights. Is it right or left multiply? Are you saying that the output scalar $y=\mathbf{w}^\top \mathbf{x}$ is a sample from the gaussian distribution? As a reader I have to read between the lines to figure out what’s the proper definition. Please write down the precise mathematical quantity.
- There is a confounding factor here regarding the idea of highly entangled neurons. If I understand properly, the output distribution of a neuron at initialization is basically a gaussian. This is because the weights are gaussian initialized unless otherwise specified. Then perhaps highly entangled neurons are basically neurons with proper feature learning, i.e. enough gradient updates, and neurons with low wasserstein distance to a gaussian could just be a neuron that didn’t receive enough gradient update. This is especially true when you consider LLMs under width scaling, as some of the parameters might not be learned properly. See https://arxiv.org/abs/2203.03466. Then it might not make sense to prune out or sparse highly entangled neurons since they could actually be the useful neurons or neurons with proper feature learning. Would like to hear what the authors think about this.

**Questions:**

See weaknesses

---

> ### Author Response · Authors · 2024-11-21
> **Official Rebuttal by Authors**
>
> Thank you for your detailed review! We have addressed each of your concerns individually, as outlined below:
>
> Weaknesses:
> 1. We would like to thank the reviewer for raising this important point. We apologize for the lack of clarity in the definition. Based on the reviewer’s suggestion, we actually revise our definition in the text for clarity. Now, neurons are defined as rows within the weight matrix (thus columns within the transpose of the weight matrix). The meaning of a neuron is still the same as we intended originally - a separate dot product across all input features of an input vector that maps to a single output feature. Thus, we now use the convention of Y = WX + b and a left-multiply of the weight matrix to the input matrix, where columns in the input matrix are individual samples. The neuron output distribution, where a neuron is a single row in W, now refers to the set of all scalar values within the corresponding row in Y, as obtained by passing all input vectors through the matrix multiply. According to the reviewer’s suggestion, we have updated the Introduction and Section 2.1 to clarify the definition of neuron and neuron output distribution.
> 2. Based on the reviewer’s note, we investigated the relationship between entanglement and training trajectory in the context of proper feature learning. For our experiments, we use the same Pythia 1.4B model that is also utilized in many other experiments throughout the paper. Pythia provides intermediate checkpoints at every 2B tokens (where each step is 2M tokens), resulting in 143 checkpoints ranging from 1,000 steps to 143,000 steps. Additionally, the initialization checkpoint (step 0) is available, making Pythia an ideal candidate for this analysis.
>
>     In our investigation, we focus on the top 3% of neurons in terms of Wasserstein distance at the last training step, the bottom 3% of neurons in terms of WD, as well as the average neuron. We tracked the characteristics of these neurons across 17 intermediate checkpoints. As predicted by the reviewer, the output distributions of all neurons were initially Gaussian at initialization. However, the highly entangled neurons began deviating from a Gaussian-like shape very early in the training process, within the first 5,000 gradient steps, as captured by their Wasserstein distance from a Gaussian distribution. In contrast, this behavior was not observed for the average neuron or the neurons with the lowest Wasserstein distances (Figure A2a).
>
>     Additionally, as a proxy for proper feature learning, when we examined the weights of each neuron across intermediate steps in terms of their divergence from the weights at initialization (Figure A2b), we found that all sets of neurons underwent changes of similar magnitudes. Interestingly, the top Wasserstein distance neurons exhibited a smaller relative change in their weights compared to the other sets. As we do not have access to more exact measures of gradient updates, we believe that the similar level of change in weights over the course of training nevertheless indicates a similar level of learning across different groups of neurons.
>
>     We appreciate the reviewer’s perspective in trying to understand entanglement from the role of LLM training, which helped us to add new insights to the paper. While we do not believe that highly entangled neurons are the only ones that have received proper feature learning, we do agree with the intuition that pruning these neurons is likely more detrimental to network performance. We plan to develop a more systematic way to prune networks that takes neuronal entanglement into account in future work. We also plan to investigate how Wasserstein neurons arise so quickly during training as well.
>
> Thank you once again for your time in reviewing our paper, we really appreciate the feedback and we hope to have addressed your concerns!

---

> > ### Comment · Reviewer_uAy8 · 2024-11-21
> > **raising score from 3 to 5**
> >
> > Thank you for the clarification on the definition of neuron output distributions and the additional experiments regarding the evolution of “Wasserstein” neurons during training. I find the results from the new experiments interesting.
> >
> > The neuron output distribution follows a gaussian distribution at initialization basically due to central limit theorem. In your experiment, the top 3% WD neurons starts to have a large WD to a gaussian very early in the training. I believe this observation can fall into a broad characterization of LLM training dynamics, i.e. gradient are low rank (https://arxiv.org/abs/1812.04754, https://arxiv.org/abs/2403.03507v2), top loss Hessian eigenvalue dynamics (https://arxiv.org/abs/2402.17457) etc. I encourage the authors to contextualize this observation in light of existing works because this WD observation might just be a different characterization of the same phenomenon during LLM or in general neural network training. My conjecture is that the low-rank gradient update induces this behavior but I’m not sure. I’m not expecting anything for this rebuttal though.
> >
> > The results on the evolution of L2 norm during training for different types of neurons mean that the Pythia 1.4B model is properly tuned in terms of all of the optimization-related hyperparameters, which is expected. However, the L2 norm of the neurons is not a good proxy for whether a neuron is “important” for downstream tasks. A neuron could have large norms but is highly redundant. I don’t think this particular experiment addresses my concern that WD neurons are potentially the only set of neurons with proper feature learning that’s aligned to downstream data.
> >
> > In general, I’m not convinced that pruning out the highly entangled neurons is a good idea. The rebuttal experiment didn’t provide sufficient evidence to counter my argument. But in light of the newly added experiment, the clarification on the definition, and the potential connections to other line of works in the field, I’m raising my score from 3 to 5. The reason I’m giving 5 but not higher is that I believe pruning out these highly entangled neurons is fundamentally not what one should do and it won’t scale, unless the authors show otherwise.

---

> ### Author Response · Authors · 2024-11-23
> **Not Pruning Entangled Neurons**
>
> Thank you for the thoughtful reply, as well as for raising your score in response to our new experiments and clarification of details.
>
> We appreciate the very pertinent references on training dynamics the reviewer has provided. We have looked through the sources, and agree that their results on how models learn quickly are in line with and provide very interesting context for the observation that Wasserstein neurons also emerge quickly. Indeed, the latter may be an emergent phenomenon as a result of the underlying training dynamics, and we have included these sources within the text. Additionally, we acknowledge the shortcoming of the $L^2$ norm experiment in demonstrating that non-Wasserstein neurons learn truly “useful” features, and have removed that claim from the text. We plan to address both of these very interesting lines of thought in future work in a setting where we can control the data flow and training more precisely.
>
> Regarding how to handle pruning entangled neurons, we actually also originally were in full agreement with the reviewer’s view that it might be better to not prune the highly entangled neurons at all. Indeed, one of our primary findings is that sparsifying Wasserstein neurons impairs the model much more than sparsifying other important neurons (Figure 3a). Furthermore, due to Wasserstein neurons not having weights that are particularly larger than average (Figure A4a), they tend to be sparsified slightly more than other neurons even by current approaches like SparseGPT (Figure 4Ab), further compounding this issue.
>
> However, we have also run a few experiments that examine the limiting case of keeping entangled neurons fully dense. For example, in Llama 2 7B for a target sparsity level $s$%, we keep the top $x$% Wasserstein neurons fully dense, and we sparsify each of the remaining neurons slightly more, to $\frac{s}{1-x}$%. We then compare this with simply pruning every neuron to $s$%, indicated in the table below as "same sparsity per neuron" (SSPN), equivalent to $x = 0$%. We use SparseGPT to sparsify the neurons, calibrated on Wikitext-2 train data and evaluated on the Wikitext-2 test set for perplexity. We test four different proportions of Wasserstein neurons to keep dense across five sparsities, as shown below. Even when just a few Wasserstein neurons are kept dense at the cost of all other neurons, performance is worse compared to when every neuron is sparsified equally. Model performance also worsens progressively as the proportion of Wasserstein neurons kept dense ($x$%) increases, since less entangled neurons are also being kept dense.
>
> ||$s=50$%|$s=60$%|$s=70$%|$s=80$%|$s=90$%|
> |:---|:---:|:---:|:---:|:---:|:---:|
> |**SSPN ($x = 0$%)**|**6.219**|**7.420**|**12.73**|**33.26**|**366.0**|
> |$x = 3$%|6.259|8.022|14.70|40.40|395.7|
> |$x = 5$%|6.345|8.131|16.03|46.67|629.5|
> |$x = 7$%|6.366|8.547|17.37|61.95|978.3|
> |$x = 10$%|6.522|9.232|19.48|79.53|8066|
>
> Thus, we find it likely that there are diminishing returns associated with keeping Wasserstein neurons fully dense. These returns are in turn outweighed by the cost of sparsifying every other sparse neuron slightly more. Therefore, we plan to investigate more advanced sparsification algorithms that account for a neuron’s degree of entanglement, but not keep them dense at the cost of every other less entangled neuron. This is why we now believe that sparsifying entangled neurons in an informed manner–rather than the two limits of either ignoring a neuron’s degree of entanglement as current approaches do, or not sparsifying entangled neurons at all as we had originally believed–will yield the best results as sparsities are pushed higher.
>
> We have added a few more details in the paper as well as this table in the appendix (Table A4) to better convey our point in this regard. Thank you once again for the very insightful comment and constructive feedback!

---

> > ### Comment · Reviewer_uAy8 · 2024-11-25
> >
> > Thank you for the additional experiments. I believe these results show that there exists a tradeoff regarding the degree of sparsifying entangled neurons, which counters my claim that we shouldn't sparsify entangled neurons at all. I would like to increase the scores to 6.
> >
> > The other thing I was thinking about is how does your approach relate to 2-bits quantizations like QTIP https://arxiv.org/abs/2406.11235. I feel like it's discussed in this paper in the form of empirical results, but it would be great to see some theoretical analyses as well. Pruning and quantization are obviously quite related, and it would be interesting to see if your approach is amenable to quantizations or how does your approach overlap with existing quantization methods in achieving model compression. This is just a generic suggestion and I don't expect the authors to incorporate discussions on quantizations during this rebuttal.

---

> > > ### Author Response · Authors · 2024-11-27
> > > **Discussion on Quantization**
> > >
> > > We would like to thank the reviewer for the additional nuances and details in our paper that their question on whether to prune entangled neurons has led to, and for their increase in score as a result.
> > >
> > > We are also very interested in investigating the new quantization techniques that have enabled quantization down to 2 bits, such as QTIP and QuIP# (https://arxiv.org/abs/2402.04396). While the focus of our work is on sparsification to high degrees, we share the view that quantizing to fewer bits is a highly complementary approach. Indeed, there is nothing theoretical that stands in the way of applying such advanced quantization techniques to each of the sparse experts in Sparse Expansion instead of the dense layers in a normal model. Doing so would also reduce the memory footprint of the experts without significantly impacting model performance, thus allowing for joint high-sparsity, low-bit compressed models.
> > >
> > > As a preliminary experiment, because GPTQ and SparseGPT share similar compression methodology, we also ran some experiments along this line of thought and tested how Sparse Expansion fares in a quantization-only setting via GPTQ. Sparse Expansion performs much better than GPTQ with fewer bits, and maintains that gap especially noticeably at 2 bits, where GPTQ suffers significant degradation. However, recently developed quantization techniques based on incoherence processing, such as QTIP, currently outperform Sparse Expansion, especially at 2 bits. Interestingly though, because Sparse Expansion allows for GPTQ to recover at 2 bits, we also speculate that it provides more performance if we use incoherence processing via QTIP as the quantization mechanism for the experts as opposed to GPTQ. If so, this may yield even more recovery at 2 bits and possibly even allow Sparse Expansion to recover model performance at 1.58 bits or 1 bit without fine-tuning. This line of thought is something that we will definitely explore in future work to further optimize Sparse Expansion for practical implementations.
> > >
> > > Table: Sparse Expansion vs. GPTQ in a quantization only setting. Llama 2 7B and Llama 2 13B were quantized to 2 and 3 bits via GPTQ or Sparse Expansion, calibrated on a subset of Wikitext-2 train data and evaluated on the Wikitext 2 test set for perplexity. While Sparse Expansion outperforms GPTQ consistently, it does not currently outperform QTIP.
> > > ||Llama 2 7B|Llama 2 13B|
> > > |:---|:---:|:---:|
> > > |Dense|5.116|4.573|
> > > |3 bit GPTQ|9.305|5.229|
> > > |3 bit Sparse Expansion|5.897|5.057|
> > > |2 bit GPTQ|427.1|22.15|
> > > |2 bit Sparse Expansion|12.06|7.387|
> > >
> > > Thanks for the thoughtful comment! While quantization is something that we do not focus on for this work, it is a major avenue in model compression, and we can explore how Sparse Expansion can be combined with complementary quantization techniques in a future project to further push model compressibility.

---

> > > > ### Comment · Reviewer_uAy8 · 2024-11-30
> > > >
> > > > Thank you for the additional experiments. This is very interesting and I hope to see more in your official release or the future project.

---

> > > > > ### Author Response · Authors · 2024-12-03
> > > > > **Thank you!**
> > > > >
> > > > > We are very excited by this avenue of research as well! We look forward to conducting further work focused on quantization, such as how Sparse Expansion can be combined with advanced incoherence processing techniques, as well as whether Wasserstein neurons play a role in quantization in addition to sparsity. Thank you once again for your very thought-provoking questions throughout the rebuttal process, we really appreciate it!

---

### Official Review · Reviewer_KmG9 · 2024-11-03

**Soundness:** 3
**Presentation:** 3
**Contribution:** 3
**Rating:** 8
**Confidence:** 3

**Summary:**

This paper examines the performance of models under weight sparsity by considering the entanglement of the neurons in the dense model. The authors first empirically show that neurons with high Wasserstein distance (WD) to a Gaussian distribution will more negatively impact the model's performance when sparsified. They then demonstrate that the WD metric has positive correlation with the Mapping Difficulty (MD) of each neuron, which serves as a heuristic for measuring neuron entanglement.

Motivated by this observation, the authors propose an algorithm (Sparse Expansion - SE) that adjusts the SparseGPT (Frantar & Alistarh, 2023) algorithm for making the model sparse: For each layer, they learn $k$ sparse weights (experts), corresponding to $k$ clusters of input data, by the SparseGPT algorithm. The clusters at each layer are identified by $k$-means clustering on the low-dimensional representations derived from PCA on the original inputs. Then, during inference, each input is routed through the closest clusters. The paper claims that each cluster will help disentangle the neurons, leading to a decrease in WD in the sparse model. Finally, they show that the sparse models learned by SE have improved performance, as measured by perplexity.

**Strengths:**

I am not fully familiar with the literature on neural disentanglement and the interpretability results. However, I find the paper’s approach to improving the performance of sparsified models by analyzing neuron interpretations interesting.

The authors take a step-by-step approach to justify their claims/arguments and to make connections between the different concepts discussed in the paper.

**Weaknesses:**

I find that many of the arguments in the paper are either based on intuition or established by showing correlations between metrics. In some cases, it is not very clear to me whether there is some sort of causation also involved. For example,  I don't understand why a neuron with a smaller WD to Gaussian should be less entangled (and thus more interpretable?). In the paper, this conclusion relies on the correlation between WD and MD, which is itself an intuitive metric for measuring entanglement. Nevertheless, the arguments are still motivating for further research in this direction.

I think the flow of the paper can be slightly improved: For example, in section 2, Wasserstein neurons and entangled neurons are used interchangeably before properly establishing their connection in section 2.3. This was a bit confusing on the first read, requiring some back and forth through the sections to clarify their connections.

(Minor) The paper's generally well-written, but it would benefit from a revision as there are several typos throughout the paper.

**Questions:**

1.  On the Sparse Expansion (SE) algorithm:
    - How is the number of clusters $k$ chosen? From Figure A2, it seems that increasing $k$ consistently improves performance (in terms of perplexity). Besides increased memory usage, are there other performance trade-offs when increasing $k$?
    - If the clustering approach mainly addresses the issue with the entangled neurons, do all neurons need the same number of clusters?
    - How much optimization/run-time overhead does the PCA+$k$-means step add to the sparseGPT algorithm when sparsifying the weights?
2.  Since the link between entanglement and Wasserstein neurons is made through the MD metric, I am wondering if we see MD also decreases by the Sparse Expansion algorithm.
3. Shouldn't the clusters learned through the SE algorithm depend on the structure of the data/task used for this purpose? For example, in Figure 10, is SE both performed and validated on Wikitext2? If so, would the sparse model still have improved performance on other downstream tasks/datasets (like the ones evaluated in Figure 2)?

---

> ### Author Response · Authors · 2024-11-21
> **Official Rebuttal by Authors (1/2)**
>
> Thank you very much for the review! We have addressed each of your concerns individually, as outlined below:
>
> Weaknesses:
>
> * We acknowledge the reviewer’s critique that we do not establish a direct causation between the metrics that we investigate. Regarding the formulation of the MD metric, prior notions of entanglement and superposition primarily operate within the space of human-interpretable features, such as using sparse autoencoders on the entire output activation to disentangle features into a much larger dimension. However, because we focus on pruning performance, we must still operate on the neuron level, information for which is lost when we move to the human-interpretable feature space. We thus derive a neuron-level metric to measure entanglement, based on principles from previous notions, for our analysis, and find that Wasserstein neurons have high MD. Unfortunately, we cannot yet make claims of direct causality, but we have also noticed that Wasserstein neurons arise fairly early on in training (Figure A2). Thus, we plan to investigate this critical training period in order to ideally establish causation in future work. Finally, in agreement with the reviewer’s perspective, we hope that this initial attempt at defining a neuronal-level metric of entanglement will inspire further research in this direction as well.
> * We would like to thank the reviewer for the recommendation to switch the section order. We agree with the assessment and have switched Sections 2.2 and 2.3 so that the terminology of and connection to entanglement is established earlier. Additionally, we have more thoroughly revised the paper for readability and typos as well.

---

> > ### Author Response · Authors · 2024-11-21
> > **Official Rebuttal by Authors (2/2)**
> >
> > Questions:
> >
> > 1. We answer the questions relating to Sparse Expansion algorithm:
> >
> >     * The reviewer pointed out correctly that increasing the number of clusters $k$ consistently improves performance but with the trade-off of increased memory footprint. Additionally, while performance improves with more clusters, the perplexity gains become marginal for larger $k$. To balance these considerations, we chose $k=16$, which provides a good trade-off between memory efficiency and performance improvement. There is also a small overhead in cluster router execution with an increasing $k$, but memory is the primary performance trade-off.
> >     * This is an excellent observation. As shown in Figure A3b not every neuron benefits the same amount for a given number of clusters, with some benefiting much more than others. This benefit is characterized by the relative improvement of Sparse Expansion with a given number of clusters over SparseGPT. Moreover, Figure A3a shows that high Wasserstein distance (WD) neurons benefit more significantly from an increased number of clusters than average. Optimizing the number of clusters for individual neurons, while promising, would require extensive engineering for efficient execution, which is beyond the scope of this work. Future versions of Sparse Expansion can incorporate optimal cluster selection based on each neuron's degree of entanglement.
> >     * Thanks for pointing this out. Based on this comment, we have now added Appendix A.5 detailing the run-time overhead of Sparse Expansion in a joint sparsification and quantization setting. We find that, with the cost of the router in place, a Sparse Expansion expert is still much faster than running a layer in a dense fashion, although it has overhead compared to a sparse linear layer without a router. Indeed, while there is still a trade-off associated with having to run multiple experts, this trade-off is very similar to that offered by standard Mixture of Experts models in terms of their strengths and weaknesses, which have become very popular recently. For pruning the model, we observe a ~$6\times$ increase in model generation time at this size, from ~$5$ minutes to ~$30$ minutes for Pythia 1.4B and from ~$4$ minutes to $25$ minutes for Llama 3.2 1B. Even though the cluster generation step is expensive during pruning, it is a one-time cost with a small overhead in inference.
> > 2. We would like to thank the reviewer for this question, and we have now run this experiment and appended this to Figure 5 (previously Figure 6). Briefly, we separate inputs and outputs into individual clusters, and within each cluster, compute the MD metric as described in Equation 2. We then average the MD across all clusters, weighted by the number of possible IO pairs within the cluster, to acquire the weighted MD. In this case, the majority (70%) of neurons do indeed experience a decrease in their weighted MD, but with a median decrease of only 2%, as opposed to the 19% for weighted WD. Neurons that had greater original MD in particular experienced a decrease in their weighted MD - within the top 10% of neurons with the highest original MD, 96% of them had a reduction in weighted MD, with a median decrease of 9%. While the reduction is not as apparent as it is for WD, MD also decreases as a result of Sparse Expansion.
> > 3. Regarding applicability to out-of-distribution (OOD) data, we agree with the reviewer that this is an important aspect to investigate. In our original setup, we used 128 samples from the Wikitext-2 train set to prepare the clustering router, pruned the individual cluster weights, and evaluated the model on the Wikitext-2 test set. Based on reviewer feedback, we have extended our evaluation to include five zero-shot benchmark tasks: ARC-Easy, ARC-Challenge, LAMBADA, SciQ, and MMLU. We conduct these experiments using two modern language models: LLaMA 3.2 1B and Pythia 1.4B. The results, shown in Table A1 and A2 in Appendix A.3, demonstrate that Sparse Expansion (with 16 clusters) outperforms all contemporary pruning algorithms across these tasks. We omit MMLU results in Pythia 1.4B because the baseline dense accuracy was less than random chance. Furthermore, all other input-aware sparsification algorithms also use calibration data from a specific dataset, so our procedure is in line with current practice.
> >
> > Thank you once again for your time in reviewing our paper, we really appreciate the feedback and we hope to have addressed your concerns!

---

> > > ### Comment · Reviewer_KmG9 · 2024-11-23
> > >
> > > Thank you for the clarifications and the additional experiments included in the paper.
> > >
> > > While I'm not deeply familiar with pruning techniques' baseline performance, I find the formulation and experiments in the paper interesting. The additional discussions added during the rebuttal can also enhance the paper's message. Thus, I’ll increase my score.

---

> > > > ### Author Response · Authors · 2024-11-24
> > > > **Thank you!**
> > > >
> > > > We sincerely appreciate the reviewer’s time and effort in reviewing our paper and would like to thank them for raising their score based on our responses and additional results. We feel that the reviewer’s insightful comments and suggestions have significantly enhanced the quality of our work. In particular, their comments prompted us to include results on out-of-distribution (OOD) data and tasks, explore different optimization strategies for the practical implementation of Sparse Expansion, and conduct experiments on the MD metric within the Sparse Expansion setting.
> > > > We are grateful for the reviewer’s thoughtful feedback. Thank you!

---

### Official Review · Reviewer_kR1B · 2024-11-04

**Soundness:** 3
**Presentation:** 3
**Contribution:** 3
**Rating:** 8
**Confidence:** 2

**Summary:**

The paper proposes using the Wasserstein distance to select entangled neurons. The authors use the mapping difficulty to measure the entanglement and argue that the Wasserstein distance is strongly correlated with it. They further introduce a sparse expansion method that disentangle the Wasserstein neurons better.

**Strengths:**

- Understanding the relationship between sparsity and entanglement is an important research area.
- The paper conducts extensive experiments to explore the connection between Wasserstein distance, entanglement, and their impact on sparsification.

**Weaknesses:**

- The concept of using the Wasserstein distance has been addressed in prior work [1]. The authors could benefit from a broader literature review. It would be helpful to discuss how other works evaluate entanglement.
- The sparse expansion approach relies on prior knowledge of the distribution to support K-means clustering. How robust is this method for out-of-distribution data?
- Practical aspects of the sparse expansion, such as runtime and inference speed (e.g., with 16 experts), should be discussed more thoroughly.
- The writing could be improved; The reviewer finds that many phrases and sentences are difficult to parse.

[1] You, Lei, and Hei Victor Cheng. "Swap: Sparse Entropic Wasserstein Regression for Robust Network Pruning." The Twelfth International Conference on Learning Representations. 2024.

**Questions:**

1. Directly normalizing the distribution to calculate the Wasserstein distance seems like a strong simplification. Is there a better approach that accounts for mean and variance?
2. In Equation (2), why is the mean used for mapping difficulty, the maximum for $N_x$ , and the median for $N_y$?
3. In Figure 4b, what does the "normalized input/output L2" represent?
4. On line 282, there is a missing word after “the.”
5. How is the weighted cluster distance calculated in Figure 6b?
6. Are the K-NN and evaluation conducted on the same training set of wikitext-2?
7. Could the authors explain why there is no correlation between the optimal number of Gaussians and relative improvement? Is this due to sparse expansion being applied directly to the weight matrix while the GMM is estimated for individual neurons? Is the relative improvement possibly influenced by the neurons with the largest GMM value? Could differences in measurement — one on the entire weight matrix, the other on individual neurons — lead to inconsistencies?
8. How do outlier weights (or neurons) overlap with neurons that have large Wasserstein distances?
9. How do neurons in the MoE model perform in terms of Wasserstein distance?

---

> ### Author Response · Authors · 2024-11-21
> **Official Rebuttal by Authors (1/3)**
>
> Thank you very much for your review! We provide individual responses to your concerns below:
>
> Weaknesses:
> 1. We would like to thank the reviewer for providing this source. We have now included this reference as well as other papers that use Wasserstein distance for the purpose of network pruning within the draft, comparing our method to theirs in the Related Work section. In terms of entanglement, most other works utilize sparse autoencoders to disentangle output activations into human-interpretable features. However, doing so loses information on the original neurons of the linear layer, while we choose to focus on the neurons due to their direct relation to network pruning. We thus derive our formulation of entanglement as an extension of currently existing ideas of superposition, while still retaining relevance to individual neurons. We have also now added additional details within the Related Work on this aspect as well.
> 2. On the point of OOD data, we have conducted additional evaluations on Sparse Expansion models that have been calibrated with Wikitext-2, but that have been evaluated on other language modeling benchmarks. We find that Sparse Expansion consistently outperforms the other sparsification algorithms of SparseGPT, Wanda, and magnitude pruning for all tested benchmarks across a variety of domains, and include this information in Appendix A.3.
> 3. Based on your suggestion, we have conducted additional experiments on the inference speed of Sparse Expansion (Table A3). We find that, with the cost of the router in place, a compressed Sparse Expansion expert is still much faster than running a layer in a dense fashion, although it has a slight overhead compared to a compressed sparse linear layer without a router. Indeed, while there is still a trade-off associated with having to run multiple experts, this trade-off is very similar to that offered by standard Mixture of Experts models in terms of their strengths and weaknesses, which have become very popular recently. Furthermore, there is significant work on efficient memory movement and offloading specifically for MoEs (Eliseev and Mazur, 2023; Kamahori et al. 2024; Xue et al. 2024; Jiang et al. 2024). Since we focus on the theoretical foundations of Wasserstein neurons and offer Sparse Expansion as an initial solution to the Wasserstein neuron problem, we do not construct a fully optimized model for inference. However, we do plan to pursue further optimizations in future works, such as the possibility of a different number of experts per neuron, as shown in Figure A3.
> 4. We appreciate this critique and have now further revised the draft for readability with this in mind.

---

> > ### Author Response · Authors · 2024-11-21
> > **Official Rebuttal by Authors (2/3)**
> >
> > Questions:
> > 1. On the topic of distribution normalization, we noticed that the neurons that benefited the most from Sparse Expansion consistently had output distributions least like a normal distribution in shape. Even neuron output distributions that had wide ranges, or were Gaussian but had nonzero mean, were much easier to sparsify than those with complex output distributions. We did consider alternative metrics to augment the normalized Wasserstein distance, but directly normalizing the output distribution as an immediate comparison of the shape across different neurons yielded the most insightful results for our studies. Thus, we chose to normalize all distributions to have zero mean and unit variance.
> > 2. In Equation 2, because the inputs to every neuron are identical, we use the maximum to scale the $L^2$ norm between pairs of inputs to be between 0 and 1 for simplicity. Regarding the output distribution, we find that normalizing the $L^2$ norm between output pairs is necessary to account for the expected range between two outputs for a given neuron. We notice that for Wasserstein neurons, their input $L^2$ vs. output $L^2$ graphs were similar to those of random neurons for a portion of the inputs, but also had a particular group of input/output pairs that had a much greater output $L^2$ for a given input $L^2$. Therefore, we use the median to avoid obfuscating this group of data, which the mean or maximum otherwise would. Finally, we use the mean of the ratios of normalized output to input $L^2$ to summarize across all pairs of inputs and outputs for each neuron for the resulting mapping difficulty.
> > 3. In Figure 2b (previously 4b), the normalized input $L^2$ refers directly to $N_x$ in Equation 2, and the normalized output $L^2$ refers directly to $N_y$ in Equation 2. This has been updated to be more clear in the text.
> > 4. We would like to thank the reviewer for this note; we have addressed this issue along with others to improve the flow and readability of our paper.
> > 5. In Figure 5b (previously 6b), to calculate the weighted cluster Wasserstein distance, the output scalar values are separated based on how their corresponding inputs were clustered. Then, within each cluster of outputs, the normalized Wasserstein distance to a Gaussian is calculated, as in Equation 1. Finally, these Wasserstein distances are averaged, weighted by the number of elements in the cluster, to acquire the weighted cluster Wasserstein distance. This has also been clarified in the caption.
> > 6. The clustering is performed with calibration data from the train data of Wikitext-2, and the evaluation is performed on the Wikitext-2 test set. However, to better investigate further OOD performance, we have also run additional benchmarks as mentioned earlier.

---

> ### Author Response · Authors · 2024-11-21
> **Official Rebuttal by Authors (3/3)**
>
> Questions:
>
> 7. We hypothesize that the optimal number of Gaussians from a GMM do not necessarily correlate relative improvement because an optimal number of Gaussians may be overfitting to the distribution in multiple cases, even though we use the Bayesian Information Criterion (BIC) metric for selecting the optimal number of components. BIC specifically penalizes an unnecessary addition of Gaussian components while trying to ensure the best fit of the model. Even for neurons with distributions resembling a Gaussian shape, a GMM might fit additional normal components to better approximate the original distribution. In contrast, the Wasserstein distance offers greater robustness in such cases, as the optimal transport distance would remain relatively low. Furthermore, while it is true that Sparse Expansion operates on the weight matrix and the optimal number of Gaussians operates on the outputs on a neuronal level, the relative improvement derived from this scheme is also measured on the order of individual neurons, and so is the Wasserstein distance of neuronal outputs. Thus, we are not fully sure that this difference can account for why the number of Gaussians does not explain relative improvement. As we were focused on accounting for the effects of high Wasserstein neurons for this work, we will further investigate why fitting GMMs do not account for relative improvement in future studies.
> 8. We investigate this phenomena and compare the average magnitude of a neuron’s weights to its output Wasserstein distance. There seems to be no positive correlation between the average weight magnitude of a neuron and its output WD. Furthermore, there is very little overlap between neurons with large mean weight magnitude and high WD. Indeed, for the top 3% of WD neurons, only one has mean weight magnitudes in the top 3% of all neurons. In addition, based on this idea, we also selectively sparsify 3% of the neurons that have the highest mean weight magnitude in the same manner as for other neurons through SparseGPT. Even in this condition, sparsifying Wasserstein neurons impairs model performance more than sparsifying neurons with the largest average weight magnitudes. We have included these results as an updated Figure 2 as well as Figure A4a.
> 9. For neurons in the Sparse Expansion MoE models, we conducted an additional experiment that tracked relative improvement as a function of the number of clusters. As the number of clusters increases, the Wasserstein neurons improve much more than low Wasserstein neurons and the average (Figure A3). Additionally, through clustering, the vast majority of neurons, and especially those with originally high WD, experience a decrease in the cluster weighted average WD (Figure 5b), the metric that your question 5 refers to. Is this what the reviewer’s question is in regard to?
>
> Thank you once again for your time in reviewing our paper, we really appreciate the feedback and we hope to have addressed your concerns!

---

> > ### Comment · Reviewer_kR1B · 2024-11-22
> >
> > The reviewer thanks the author for their detailed responses and for including the OOD results and other investigations. I encourage the author to incorporate some of these discussions into the appendix—for example, the discussion related to Question 2.
> >
> > I have raised my score from 6 to 8. However, I would like to leave a note for the Area Chair that I am not an expert in interpretability and explainable AI, and therefore, I will maintain my current confidence score.

---

> > > ### Author Response · Authors · 2024-11-23
> > > **Thank you!**
> > >
> > > We would like to thank the reviewer for their very helpful feedback throughout the rebuttal process and for raising their score based on our responses and additional results. We have also followed the suggestion to incorporate the discussion related to Question 2 into the appendix to provide more clarity on how the MD metric was derived (Appendix A.8). By addressing the very thoughtful points that were previously brought up, we feel that the paper is now much stronger than it was before, and we are very appreciative of the constructive critiques by the reviewer!

---

### Meta-Review · Area_Chair_PgyU · 2024-12-20

**Metareview:**

The paper explores neuron entanglement in sparse neural networks, proposing that a high Wasserstein distance (WD) to a Gaussian indicates entangled, polysemantic neurons whose removal harms performance. A sparse expansion method is introduced to improve performance by clustering inputs and training sparse "experts" for each cluster, theoretically disentangling neurons.

The reviewers noted a lack of novelty of using Wasserstein distance, and raised concerns about some assumptions and computational costs. Nevertheless, the strong experimental analysis and promising results will be of interest to the community and I recommend acceptance.

**Additional Comments On Reviewer Discussion:**

The reviewers raised numerous questions in the discussion which were largely answered to the reviewers' satisfaction during the discussion phase. The authors defended their methodological choices, provided additional data and clarifications on the Sparse Expansion method and its evaluation, and addressed specific questions about the MoE implementation. They acknowledged some open questions for future research, particularly regarding the GMM analysis. The authors also provided some additional experimental analysis on sparsifying entangled neurons, which eased some concerns.

---

### Decision · Program_Chairs · 2025-01-22

Accept (Spotlight)